# EGCG binds intrinsically disordered N-terminal domain of p53 and disrupts p53-MDM2 interaction

Jing Zhao[1,2,10], Alan Blayney [3,10], Xiaorong Liu[4], Lauren Gandy[2], Weihua Jin[2], Lufeng Yan[2], Jeung-Hoi Ha[3], Ashley J. Canning[3], Michael Connelly [3], Chao Yang[5], Xinyue Liu[2], Yuanyuan Xiao[2], Michael S. Cosgrove[3], Sozanne R. Solmaz[6], Yingkai Zhang [5,7], David Ban[8], Jianhan Chen [4,9], Stewart N. Loh[3] & Chunyu Wang [2✉]

Epigallocatechin gallate (EGCG) from green tea can induce apoptosis in cancerous cells, but the underlying molecular mechanisms remain poorly understood. Using SPR and NMR, here we report a direct, μM interaction between EGCG and the tumor suppressor p53 ($K_D = 1.6 \pm 1.4$ μM), with the disordered N-terminal domain (NTD) identified as the major binding site ($K_D = 4 \pm 2$ μM). Large scale atomistic simulations (>100 μs), SAXS and AUC demonstrate that EGCG-NTD interaction is dynamic and EGCG causes the emergence of a subpopulation of compact bound conformations. The EGCG-p53 interaction disrupts p53 interaction with its regulatory E3 ligase MDM2 and inhibits ubiquitination of p53 by MDM2 in an in vitro ubiquitination assay, likely stabilizing p53 for anti-tumor activity. Our work provides insights into the mechanisms for EGCG's anticancer activity and identifies p53 NTD as a target for cancer drug discovery through dynamic interactions with small molecules.

[1] College of Food Science and Nutritional Engineering, China Agricultural University, Beijing, China. [2] Center for Biotechnology and Interdisciplinary Studies, Department of Chemistry and Chemical Biology, Department of Biological Sciences, Rensselaer Polytechnic Institute, Troy, NY, USA. [3] Department of Biochemistry and Molecular Biology, SUNY Upstate Medical University, Syracuse, NY, USA. [4] Department of Chemistry, University of Massachusetts, Amherst, MA, USA. [5] Department of Chemistry, New York University, New York, NY, USA. [6] Department of Chemistry, State University of New York at Binghamton, Binghamton, NY, USA. [7] NYU-ECNU Center for Computational Chemistry at NYU Shanghai, Shanghai, China. [8] Merck Research Laboratories, Mass Spectrometry and Biophysics, Kenilworth, NJ, USA. [9] Department of Biochemistry and Molecular Biology, University of Massachusetts, Amherst, MA, USA. [10] These authors contributed equally: Jing Zhao, Alan Blayney. ✉email: wangc5@rpi.edu

Diet-based cancer prevention and therapy have received considerable attention in recent years. Green tea, a popular beverage consumed worldwide, has been reported to have inhibitory effects against various types of cancer, such as breast, lung, prostate, and colon cancer. Most of the chemopreventive effects of green tea on cancer are attributed to polyphenol compounds, among which epigallocatechin-3-gallate (EGCG) is the most important[1]. EGCG accounts for 50–80% of the catechin in green tea. There is 200–300 mg of EGCG in a brewed cup (240 mL) of green tea[2]. By drinking cups of green tea or taking an EGCG tablet, a serum concentration of 0.1–1 μM EGCG can be achieved[3,4]. The anti-cancer effect of EGCG has been demonstrated in epidemiological, cell culture, and animal studies, and in clinical trials[5]. A 10-year prospective study by Nakachi and Imai reported a decreased risk of cancer for those consuming over 10 cups of green tea a day, compared with those consuming below three cups[6,7]. Recently, Shin et al. found that green tea extract reduced the recurrence rate of colorectal adenomas by 44.2% in a randomized clinical trial in Korea[8]. In vitro, EGCG was shown to promote cell growth arrest and induce apoptosis in a variety of human cancer cell lines, including prostate carcinoma cells[9,10], epidermoid carcinoma cells[11], bladder cancer cells[12], and colon cancer cells[13]. In vivo, oral or intravenous administration of green tea or purified EGCG in mice inhibited angiogenesis and restrained solid tumor growth[14,15]. At the molecular level, EGCG has been demonstrated to interact with cancer-related proteins, such as glucose-regulated protein 78 (GRP78)[16] and Ras–GTPase-activating protein SH3 domain-binding protein 1 (G3BP1)[17], with approximately μM affinities.

In EGCG-induced apoptosis and cell growth arrest, p53 was found to play an important role[18,19]. p53, often referred to as "the guardian of the genome", is a crucial tumor suppressor mutated in over 50% of human cancer. p53 promotes cell-cycle arrest or apoptosis as a response to cellular stress stimuli, such as oxidative stress, oncogene activation, and DNA damage[20,21]. As a transcription factor, p53 is tightly regulated with a short half-life. p53 protein is normally maintained at low levels in healthy mammalian cells by continuous ubiquitylation and subsequent degradation, mediated by murine double minute 2 (MDM2) E3 ligase. Under cellular stress, ubiquitylation of p53 is suppressed and p53 is stabilized. p53 then accumulates in the nucleus and turns on expression of target genes, triggering cell-cycle arrest, apoptosis, and DNA-repair processes[20]. Besides acting as a transcription factor, p53 can also translocate to the cytoplasm or mitochondria. p53 interacts directly with anti-apoptotic proteins such as Bax and Bcl2 to induce apoptosis[21,22] and is also involved in the anti-senescent effect of EGCG[23,24].

Full length p53 is composed of an N-terminal domain (NTD), a DNA-binding domain (DBD), a tetramerization domain (TET), and a C-terminal regulatory domain (REG) (Fig. 1). The NTD is further divided into two transcriptional activation domains (TAD1 and TAD2) and a proline-rich domain (PRD). NTD is an intrinsically disordered protein (IDP) and interacts with many proteins, acting as a hub for cellular signaling[25,26]. NTD is not only required for transactivation, but also binds MDM2 to mediate the ubiquitylation and degradation of p53. Independent of ubiquitylation, MDM2 also inhibits transcription by preventing general transcription factors from binding to NTD[27]. The apoptosis effect of EGCG on human cancer cells was associated with its interference of MDM2-mediated p53 ubiquitylation[28]. EGCG is also reported to stabilize p53, with increased phosphorylation on critical serine residues[29]. In a recent study, EGCG was identified from a library of 2295 phytochemicals as an inhibitor of p53–MDM2 interaction[30]. However, the molecular mechanism of how EGCG disrupts MDM2–p53 interaction is not yet understood.

In this work, we demonstrate the direct binding between EGCG and p53, mediated by NTD of p53. We show that the EGCG–p53 interaction disrupts p53 interaction with MDM2 and inhibits ubiquitination of p53, likely stabilizing p53 for anti-tumor activity, providing a structural mechanism for the anticancer effect of EGCG.

## Results

**SPR shows similar μM affinity for EGCG binding to full-length p53 and NTD of p53.** EGCG–p53 interaction was studied using a sensor chip immobilized with full-length p53 or p53 NTD. As shown in Fig. 1a, b, specific binding of EGCG to full-length p53 and p53 NTD were detected by SPR. The dissociation constant ($K_D = 4 \pm 2$ μM) for p53 NTD and EGCG interaction was similar to that ($K_D = 1.6 \pm 1.4$ μM) for full-length p53 and EGCG interaction. The two dissociation constants are within measurement error, indicating that NTD is the major mediator of the p53–EGCG interaction. NTD is also one of the primary binding sites for MDM2[31], suggesting this EGCG–NTD interaction may impact MDM2-mediated p53 degradation and inhibit NTD interactions with general transcription factors.

**Mapping the binding epitopes within EGCG by saturation transfer difference (STD) NMR.** As a first step towards understanding the structural mechanism of EGCG–p53 interaction, STD NMR was employed to map the binding epitopes within EGCG to both full-length p53 and p53 NTD. In STD NMR, ligand protons at the ligand–protein interface show strong signals, enabling the mapping of the ligand's binding epitope[32]. The assignment of EGCG was accomplished by a series of 2D experiment including $^1$H–$^1$H COSY, $^1$H–$^1$H TOCSY, $^1$H–$^{13}$C HSQC, $^1$H–$^{13}$C HMQC, and $^1$H–$^1$H NOESY. The STD and reference spectrum (STR) obtained for EGCG–p53 mixture are shown in Fig. 1c, d. An STD spectrum recorded for EGCG in absence of p53 was employed as a negative control, where no STD signal was detected as expected. Overall similar STD signals of EGCG were observed for p53 NTD and full-length p53, consistent with the similar binding affinity detected by SPR, again indicating that NTD is the major binding site for p53–EGCG interaction. To elucidate the binding epitopes in EGCG, a normalized STD amplification factor ($STD_{af}$) was calculated with the highest $STD_{af}$ set to 100%[33]. Protons with large $STD_{af}$ are distributed throughout the EGCG molecule (Fig. 1e), indicating that most parts of EGCG molecule are involved in NTD interaction. However, phenol hydroxyl resonances cannot be detected in solution NMR due to fast solvent exchange; thus the role of these hydroxyl group cannot be established based on STD data.

**Mapping EGCG-binding site on p53 NTD by 2D NMR titration.** To map the binding site of EGCG on NTD, a series of 2D $^{15}$N–$^{13}$C NCO (amide nitrogen–carbonyl correlation) experiment were recorded. In NCO, $^{13}$C dimension and $^{15}$N dimension represent the carbonyl carbon (C', of residue $i-1$) and amide nitrogen (N$^H$, of residue $i$) of a peptide bond, respectively. The assignment of NCO was based on Wong et al.[34], and confirmed by 3D HNCO, HNCACB, (H)CANCO, and (H)CANCO-I. The chemical shift perturbations (CSPs) in NCO spectra upon EGCG binding are the most prominent in two areas: W23-K24 and P47-T55, as shown in Fig. 2b. Residues with the largest CSPs were highlighted in Fig. 2a. Two tryptophan residues, W23 and W53 (colored red in the NTD sequence), produced the most positive $\Delta\delta^{C'}$, indicating that EGCG binding increased C' chemical shift of W23 and W53. Several nearby residues, F54, Q52, and T55, also displayed highly positive $\Delta\delta^{C'}$. CSPs of backbone $^{15}$N ($\Delta\delta^N$) were largest at W23-K24 and D48-F54, which is similar to the CSPs in

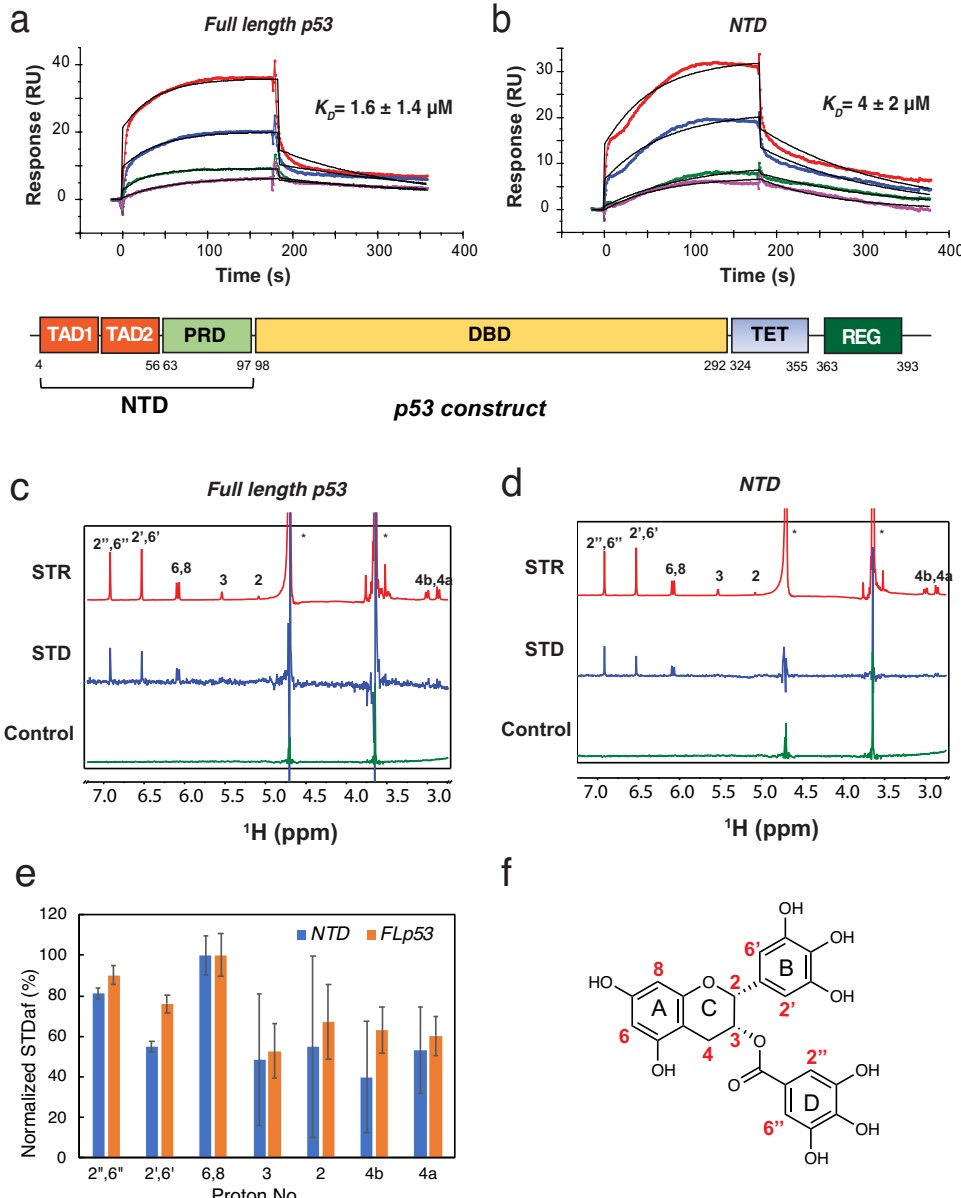

**Fig. 1 EGCG binds full-length p53 and NTD with similar affinity and binding sites, determined by SPR and STD NMR. a** SPR sensorgrams and binding affinity of full length p53–EGCG interaction. Full-length p53 was immobilized on the sensor chip and the concentrations of EGCG (from top to bottom) were 10, 5, 1, and 0.5 μM, respectively. The black curves are the fit curves using models from BIAevaluation 4.0.1. Chi$^2$ (a measure of the average deviation of the experimental data from the fitted curve) = 2.0. **b** SPR sensorgrams and binding affinity of p53 NTD–EGCG interaction. P53 NTD was immobilized on the sensor chip and the concentrations of the EGCG (from top to bottom) were 10, 5, 1, and 0.5 μM, respectively. Chi$^2$ = 0.7. Domain organization of p53 is shown below the SPR sensorgrams. **c** Saturation transfer reference (STR) (red) and saturation transfer difference (STD) (blue) spectra of 1 mM EGCG in the presence of 10 μM p53 NTD, and negative control (green, STD NMR of EGCG without NTD protein). **d** STR (red) and STD (blue) spectra of 1 mM EGCG in the presence of 5 μM full-length p53. Asterisks denote peaks due to $H_2O$ ($\delta = 4.7$ ppm) and tris (hydroxymethyl) aminomethane ($\delta = 3.7$ ppm) buffer. **e** Bar graph of normalized STD amplification factor (STD$_{af}$) plotted against assigned protons on EGCG. Data are presented as the average of multiple scans (n = 256) and error bars represent standard error estimated by the signal to noise ratio. Source data are provided as a Source Data file. **f** Chemial structure of EGCG and proton numbering.

the $^{13}C$ dimension. Such a combination of positive $\Delta\delta^{C'}$ and negative $\Delta\delta^{N}$ suggests a coil to helical conformational transition[35,36]. Interestingly, EGCG-binding sites near W23 and W53 overlap with the regions with largest α-helical propensity in apo NTD (Fig. 2c), as determined by the program secondary structure propensity (SSP)[37]. In contrast, the C-terminal proline-rich region showed smaller and both negative $\Delta\delta^{C'}$ and $\Delta\delta^{N}$, which is likely due to hydrogen bonds formation between EGCG phenolic hydroxyl and proline carbonyl[38,39].

**EGCG induces conformational heterogeneity in NTD by small-angle X-ray scattering (SAXS), AUC, and MD simulation.** SAXS experiments were carried out to probe conformational changes in the NTD induced by EGCG. Here we used a shorter construct NTD 20–70 which includes the major binding sites of EGCG. Results from dynamic light scattering (DLS) and Guinier plots of the SAXS data suggest that the NTD is monodisperse, with a radius of gyration of 23.9 Å (Supplementary Figs. 1–3, Supplementary Table 1, see "Methods" section). The molar mass

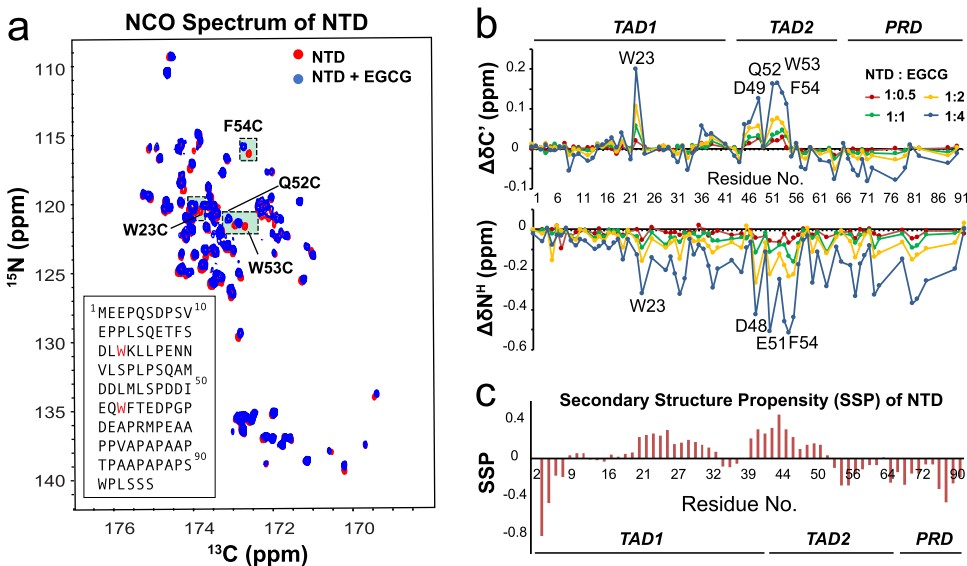

**Fig. 2 $^{15}$N–$^{13}$C NCO titration of p53 NTD with EGCG. a** Overlay of $^{15}$N–$^{13}$C NCO spectra of 0.3 mM p53 NTD, apo (red) and in the presence 1.2 mM EGCG (blue). **b** CSPs of carbonyl carbon (C′) and amide nitrogen (N$^H$) of p53 NTD (0.3 mM) titrated with EGCG, at NTD:EGCG ratios of 1:0.5 (red), 1:1 (green), 1:2 (yellow), and 1:4 (blue). Residues with largest CSPs were marked by residue type and number. Source data are provided as a Source Data file. **c** Secondary structure propensity (SSP) of p53 NTD. An SSP score of 1 or −1 reflects fully formed α-helix or β-sheet structure, while a score of ~0.5 or ~−0.5 indicates significant propensity for α-helix or β-sheet, respectively. Source data are provided as a Source Data file.

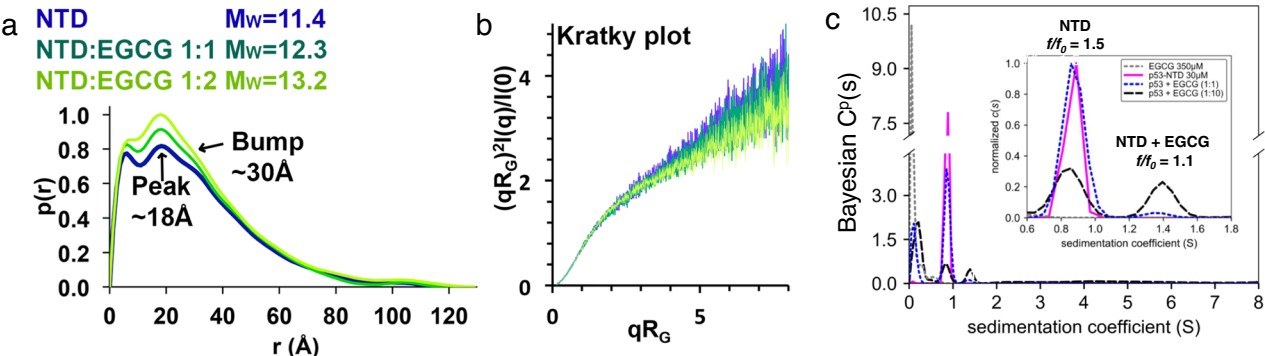

**Fig. 3 SAXS and SV-AUC experiments showed that EGCG induces conformational change and the emergence of a compact subpopulation of NTD. a** Pair distance distribution ($p(r)$) functions were derived from the scattering intensity profiles (Supplementary Figs. 1–3). The $p(r)$ functions of the NTD in the absence and presence of EGCG at molar ratios of 1:1 and 1:2, normalized to the highest signal of the NTD/EGCG 1:2 plot, are shown. Molar masses MW were determined with the 'Size & Shape' method[71]. **b** Dimensionless Kratky plots of the SAXS data (see **a** for color code; $q$: scattering vector, $R_G$: radius of gyration, $I(q)$: scattering intensity). For more detailed SAXS methods and Guinier plots, see Supplementary Table 1 and Supplementary Figs. 1–3. **c** EGCG induces the emergence of compact EGCG-bound NTD conformation. Diffusion deconvoluted sedimentation coefficient distributions $c(s)$ of 300 μM EGCG (gray dashed line) and 30 μM p53-NTD (solid pink line). The $s$ values derived from this analysis were used as prior expectations in Bayesian analyses ($c^P(s)$) of SV-AUC data of P53-NTD in the presence of 30 μM EGCG (blue dashed line), or 300 μM EGCG (black dashed line). The distributions were normalized by total integrated area. The inset shows a blow-up of the region between 0.6$s$ and 1.8$s$ normalized by the highest peak in the distribution.

derived from the SAXS data showed that the apo NTD 20–70 is a monomer (MW = 6.4 kDa; while the expected MW based on sequence is 6.9 kDa). In pair distance distribution functions ($p(r)$) derived from SAXS profiles (Fig. 3a), the profile of apo NTD displayed two peaks at distances of ~6 and 18 Å and a small bump at ~30 Å, suggesting an extended and elongated conformation that is divided into subdomains. Addition of EGCG (1:1 and 1:2) resulted in an increase of the relative height of the peak at a distance of 18 Å and led to the decrease of the small bump at 30 Å, suggesting a shift of the conformational equilibria induced by EGCG binding. The normalized Kratky plots of the NTD show that upon addition of EGCG, the signal is modestly reduced at high $q$ values in a concentration-dependent manner (Fig. 3b).

These results suggest that binding of EGCG to the NTD either causes a subtle conformational change in the entire population of NTD, or that it induces a sub-population of a more compact form that cannot be resolved from the disordered sub-population within the signal to noise of the SAXS experiment.

To distinguish these two possibilities, we performed sedimentation velocity analytical ultracentrifugation (SV-AUC) experiments on four samples: EGCG, NTD, EGCG + NTD at a 1:1 molar ratio, and EGCG + NTD at a 10:1 ratio (Fig. 3c, Supplementary Figs. 4, 5). The resulting continuous sedimentation coefficient distribution analyses ($c(s)$) showed that the peak corresponding to NTD broadened and shifted to a larger $s$-value in an EGCG concentration-dependent manner (Supplementary

Fig. 5). With a 10-fold excess of EGCG, the peak corresponding to NTD is significantly broadened and shifted to 1.1$s$, which likely reflects a reaction boundary between bound and unbound species that cannot be resolved within the signal to noise of the experiment using the standard maximum entropy (ME) regularization routine in SEDFIT[40]. Hypotheses include rapid inter-conversion between a disordered and more compact form of NTD, or between monomeric and oligomeric NTD species. To distinguish these hypotheses, we reanalyzed the SV-AUC data using a Bayesian approach, which takes advantage of available prior information in the ME regularization to assign different probabilities to different $s$-value regions in a distribution[41]. Using the experimentally determined $s$-values of EGCG and apo-NTD as prior expectations, the resulting Bayesian sedimentation coefficient distributions (Bayesian $c^{(p)}(s)$) (Fig. 3c) showed peaks for EGCG at ~0.2$s$ (gray dashed curve), apo-NTD at ~0.8$s$ (pink solid curve), and a new peak at ~1.4$s$ that was dependent on the concentration of EGCG and not explained by the prior expectations (dashed blue and black curves). The experimentally determined frictional coefficient ($f/f_0$) of apo-NTD at 1.5 is consistent with an elongated or intrinsically disordered protein with a mass estimate of 6.9 kDa, which is in excellent agreement with the monomer mass of NTD. In contrast, the species giving rise to the peak at 1.4$s$ gave a frictional ratio of ~1.1 based on the expected mass of a 1:1 complex between NTD and EGCG of ~7.4 kDa. Dimerization of NTD can be ruled out, as simulation using SEDNTERP[42] reveal that a particle with the mass of a dimer is expected to sediment at 1.9$s$. Thus, the 1.4$s$ species corresponds to a compact conformation of NTD. EGCG-binding induces the emergence of a subpopulation of compact NTD conformation. In EGCG-bound state, NTD is rapidly interconverting between disordered and compact conformations.

To further understand the details of EGCG/NTD interaction and to validate the results from NMR, SAXS, and AUC, we carried out atomistic molecular dynamics (MD) simulations in explicit solvent using an enhanced sampling technique known as replica exchange with solute tempering (REST)[43,44]. The recently optimized a99SB-disp force field[45] was used, which proves to be particularly suitable for IDPs and can accurately reproduce a set of nontrivial local and long-range structural properties of p53–NTD (1–61)[46]. Each REST2 simulation consisted of 16 replicas with effective solute temperatures spaced exponentially between 298 and 500 K (Supplementary Fig. 7 and see "Methods" section for more details). Total simulation times for NTD and NTD/EGCG simulations were 3.4 and 3.9 μs/replica for an aggregated sampling time of ~120 μs, making this one of the most extensive explicit solvent simulations of p53–NTD. The deployment of enhanced sampling as well as the length of the simulations are crucial to achieve adequate level of convergence in the simulated ensembles for resolving the effects of EGCG binding on p53–NTD and the nature of their interactions. The presence of EGCG perturbs the overall conformation of NTD. As shown in the pairwise distance distribution function $P(r)$, calculated from MD simulations, EGCG caused a significant increase in the probability at ~15 Å and a decrease in probability at larger distances (Fig. 4a). This is in qualitative agreement with the $P(r)$ derived from SAXS, which also shows an increased distribution of pairwise distances at 15–18 Å upon addition of EGCG (Figs. 3a, 4a and Supplementary Figs. 4, 5, 8a, 10). The decrease of the bump at ~30 Å in $P(r)$ curve from SAXS is also mirrored in the overall decrease of $P(r)$ values at above 30 Å from MD. The conformational changes of NTD induced by EGCG binding can be further visualized using PCA analysis (see SI for details). As shown in Fig. 4c, the presence of EGCG can significantly shift NTD conformational equilibria, leading to more conformational heterogeneity and the emergence of a

subpopulation of more compact conformations (also see Supplementary Fig. 8a). This is consistent with the results from AUC and Bayesian analysis of the AUC data (Fig. 3). The emergence of the compact conformation apparently leads to increased probabilities of shorter pairwise distances (e.g., 15–18 Å) in Fig. 4a. In addition, EGCG also appears to increase helical propensities of NTD (Supplementary Fig. 9).

Atomistic simulations further reveal that the EGCG/NTD interaction is highly dynamic. Many p53–NTD residues show significant probabilities of engaging EGCG (Fig. 4b). This is consistent with NMR titration experiments showing broad chemical shift perturbations. Interestingly, the same putative EGCG-binding sites identified with NMR (Fig. 2b), W23-K24 and P47-T55, also showed elevated probabilities of contacting EGCG in the simulated ensembles (Fig. 4b). Principle component analysis (PCA), based on heavy-atom distances between NTD and EGCG, resolved a large number of possible NTD/EGCG complex structures (Fig. 4d). EGCG binding caused large increase in $^{15}N$ transverse relaxation in NTD (Supplementary Fig. 11), likely caused by chemical exchange between multiple EGCG-bound states. This is consistent with the multiple bound conformations observed in MD simulation of EGCG-bound state. In contrast to specific interactions that are often required to achieve high binding affinity between small molecules and folded proteins, such dynamic interfaces between NTD and EGCG represented a unique binding mechanism that small molecules could utilize for tight interaction with IDPs by retaining high entropy.

Taken together, the NMR, MD, SAXS, and SV-AUC results show that EGCG binding induces conformational heterogeneity and the emergence of a subpopulation of compact conformation in NTD.

**EGCG disrupts p53–MDM2 interaction and inhibits MDM2-mediated p53 ubiquitination in vitro.** EGCG has been reported to disrupt the MDM2 and p53 interaction in human lung cancer cells, which resulted in the inhibition of MDM2-mediated p53 ubiquitylation and subsequent degradation[31]. However, the mechanism of the disruption is not well understood, e.g. whether EGCG directly disrupt the MDM2–p53 complex or EGCG acts through an indirect mechanism. Because EGCG binds to NTD, the site for p53–MDM2 interaction, we tested whether EGCG can directly disrupt p53–MDM2 interaction. The interaction of p53–MDM2 relies on the shape complementarity between a MDM2 cleft and the hydrophobic side of an α-helix in NTD, in which F19, W23, and L26 insert deeply into a cleft on the surface of MDM2[32]. This tight-fitting interaction may be disrupted by EGCG binding directly to the W23 region and/or the more global conformational change and shift in NTD caused by EGCG. To characterize the inhibitory effect of EGCG on p53–MDM2 interaction, p53–MDM2 binding was first measured by SPR with MDM2 immobilized on the sensor chip. As shown in Fig. 5a, different concentrations of full-length p53, from 0.125 to 2 μM, were flowed over the chip surface, resulting in five binding curves fitted with a $K_D$ value of 0.6 ± 0.1 μM, similar to the affinity (~0.3 μM) from the literature[26,47]. To detect the inhibition of EGCG on p53–MDM2 interaction, different concentrations of EGCG were pre-incubated with p53 and then flowed over the MDM2-immobilized chip surface, as illustrated in Fig. 5c. Sensorgrams of protein–ligand mixture with different EGCG concentration were stacked in Fig. 5b. With the increase of EGCG concentration in the solution, the SPR signal decreased, indicating reduced p53–MDM2 interaction. 500 nM of EGCG eliminated most of the binding between p53 and MDM2. An IC$_{50}$ of around 0.2 μM was obtained, close to the binding affinity of p53 and MDM2

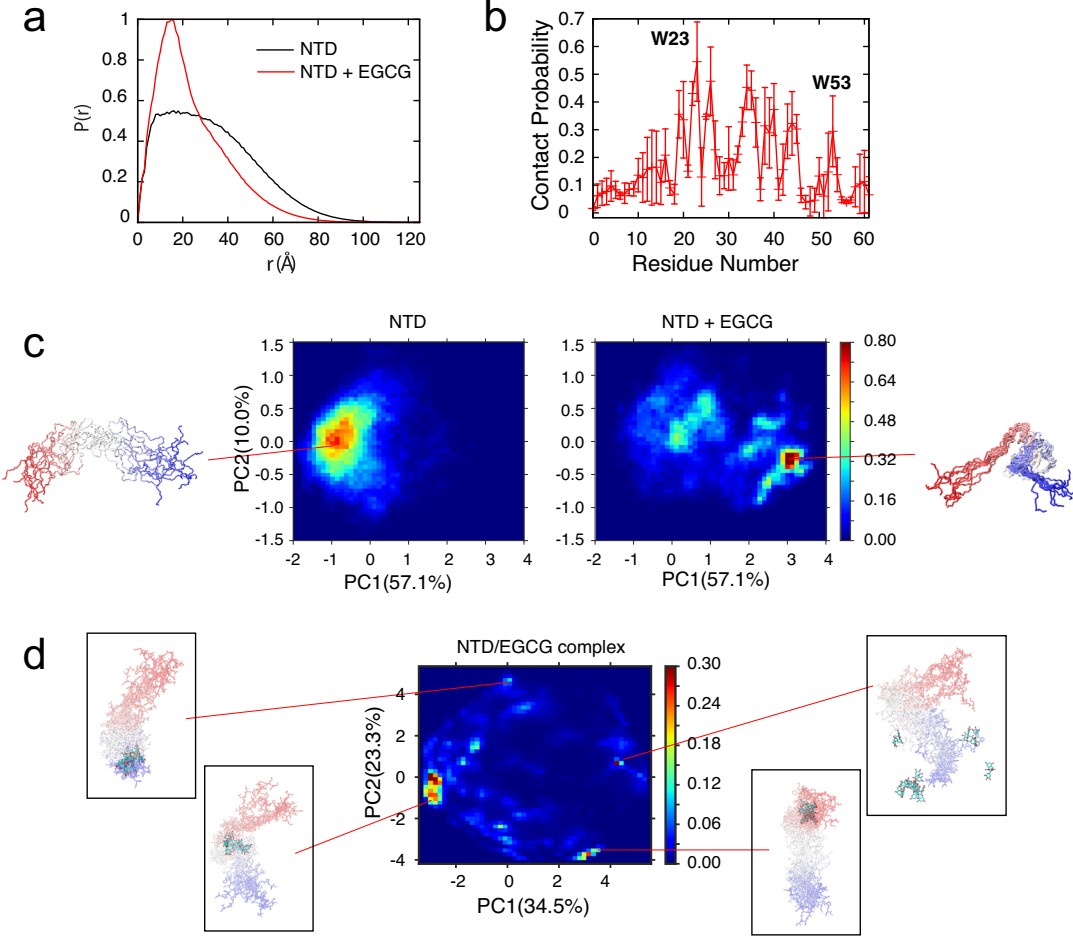

**Fig. 4 EGCG-induced conformational changes of NTD and EGCG–NTD complexes derived from atomistic simulations. a** Pairwise distance distributions in the presence or absence of EGCG molecules calculated from REST2 simulations. **b** Probability of contacts between EGCG and each residue of p53–NTD. A contact was defined as those with minimal heavy atom distance ≤4.2 Å. The contact probability was presented as the mean and standard deviation (error bars) of three subtrajectories ($n = 3$). Source data are provided as a Source Data file. **c** Probability distribution of simulated disordered ensembles of NTD projected onto the first two principal components (PCs) in PCA analysis in the absence (left) or presence (right) of EGCG. Values in the parenthesis are percentages of variance along each direction. Ten representative structures for the most populated basin of each state are shown, with a color gradient from red at the N-terminus to blue at the C-terminus. **d** Probability distribution of the NTD/EGCG complex structures projected onto the first two PCs in PCA analysis. Representative structures for a few highly populated basins are shown, with the bound EGCG represented in licorice.

(0.6 μM). We further validated the inhibition effect of EGCG by fluorescence anisotropy with nutlin as a positive control, using full-length, N-terminally labeled fluorescein–MDM2 binding to full-length p53 (Supplementary Fig. 12). In these experiments, an excess of EGCG (25 μM) competed with the protein–protein interaction, weakening the apparent $K_D$ between p53 and MDM2 in solution from 2.4 to 340 nM. The commercially available MDM2-inhibitor Nutlin-3a[48] was used as a positive control; 1 μM Nutlin-3a was found to weaken the apparent $K_D$ to 37 nM (Supplementary Fig. 12).

MDM2 is the primary ubiquitin ligase acting on p53, targeting it to the proteasome for degradation. To determine whether EGCG's inhibition of p53–MDM2 interaction disrupts this process, we performed in vitro ubiquitination experiments using full length p53 and MDM2, as well as the requisite E1 and E2 enzymes. With excess ubiquitin, p53 is efficiently multi- and poly-ubiquitinated, producing conjugates of varying molecular weights (lane 2 of Fig. 6a, Supplementary Fig. 13a). The addition of high concentration of EGCG completely stops this process (lane 3), while the addition of lower concentrations produces a dose–response trend (lanes 4–10). By using fluorescein-conjugated p53, the extent of ubiquitination can be quantified

via fluorescence densitometry (Supplementary Fig. 13c). Due to the multi-step nature of this process, the kinetics of formation of the different ubiquitinated p53 products are complex (Supplementary Fig. 13d); however, the conjugation of even a single ubiquitin is sufficient to alter p53 function. We therefore track the level of unmodified p53, and find that EGCG has an IC50 of ~100 μM under the experimental conditions (Fig. 6b). Importantly, this IC50 value describes inhibition of an enzymatic process, and is entirely distinct from p53–EGCG binding by SPR.

## Discussion

Our data showed that EGCG disrupts p53-MDM2 interaction and inhibits the ubiquitination of p53 mediated by MDM2, a major degradation pathway for p53. These results provide more details to the mechanism of EGCG-induced apoptosis in cancer cells and the efficacy of EGCG in cancer prevention and possible treatment. In addition, p53 NTD not only contains the transactivation domain important for regulating genes expression[49,50] and for response to DNA damage[51], but is also involved in many cellular signaling pathway by binding to different proteins, such as prolyl isomerase Pin1[52] and focal adhesion kinase[53]. Therefore,

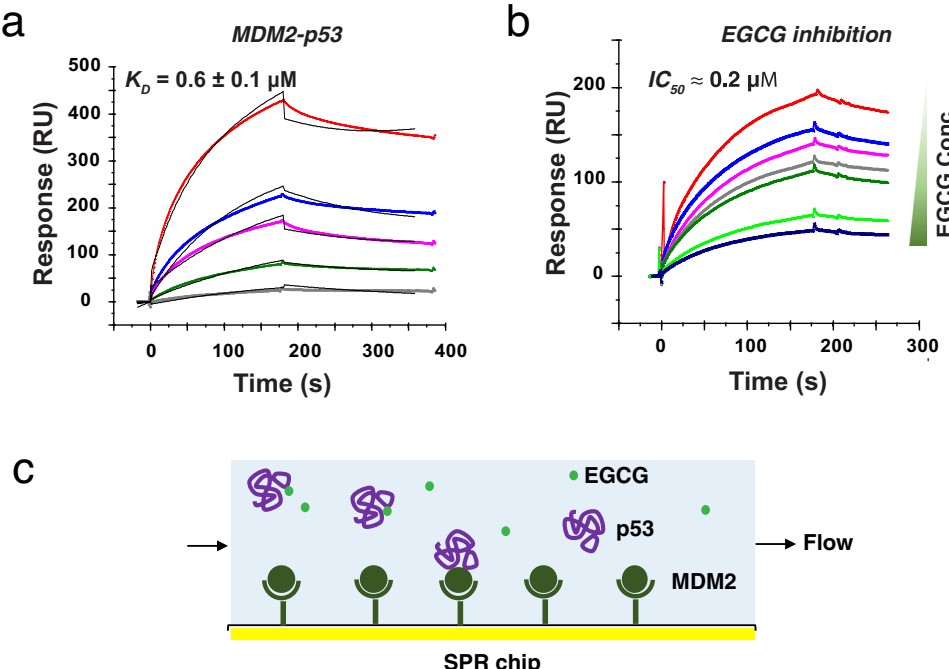

**Fig. 5 EGCG inhibits MDM2–p53 interaction with a sub-micromolar IC$_{50}$. a** SPR sensorgrams and binding affinity of full length p53–MDM2 interaction. Concentrations of the full-length p53 (from top to bottom) were 2000, 1000, 500, 250, and 125 nM, respectively. The black curves are the fit curves using models from BIAevaluation 4.0.1. Chi$^2$ = 31. **b** SPR sensorgrams of full length p53–MDM2 binding in the presence of different concentrations of EGCG in solution. Concentrations of EGCG (from top to bottom) were 0, 31.25, 62.5, 125, 150, 300, and 500 nM, respectively, while concentration of p53 remained at 250 nM. Source data are provided as a Source Data file. **c** Experimental setup for SPR competition. When EGCG binds p53 in solution, less p53 is available to bind MDM2 on the chip, reducing SPR signal.

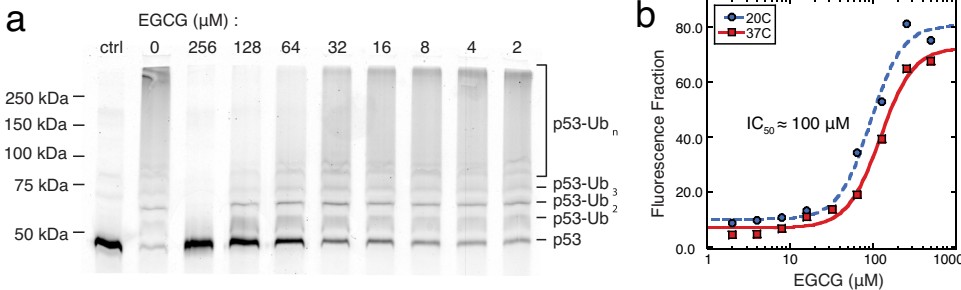

**Fig. 6 EGCG inhibits the MDM2-catalyzed ubiquitination of p53 in vitro. a** In vitro ubiquitination of fluorescein-tagged p53 was carried out in the presence of up to 0.25 mM EGCG at 20 °C. The results were visualized by fluorescence SDS–PAGE. **b** Plotting the fluorescence fraction of residual unmodified p53 as a function of EGCG concentration and fitting to a four-parameter sigmoid indicates that EGCG has an IC$_{50}$ of ~100 µM for the MDM2-catalyzed ubiquitination of p53.

EGCG–p53 NTD interaction may have wide-ranging implications by modulating the stability and binding properties of p53.

IDPs such as NTD are notoriously challenging targets for drug discovery. Here we show that a small molecule, EGCG with MW of 458 Da, can efficiently disrupt an important physiological interaction between p53 and MDM2, with a dynamic interface. Previous drug discovery aimed at disrupting p53–MDM2 interface focused on the NTD-binding pocket on MDM2 surface. Our data demonstrate that even IDPs, such as NTD of p53, may be good targets for small molecule drug discovery[54,55].

In summary, our work demonstrated that EGCG binds directly to tumor suppressor p53. Using SPR and NMR, we identified the N-terminal domain (NTD) of p53 as the major EGCG-binding site, which coincides with the MDM2-binding site. Combining NMR, atomistic simulation, AUC, and SAXS, we showed that EGCG binding has multiple binding modes, and induces conformational heterogeneity and the emergence of a compact conformation in NTD. EGCG may thus shield the MDM2-binding site of p53–NTD to inhibit p53 degradation. Indeed, we demonstrated that EGCG disrupts the binding of p53 to its regulator MDM2, thus stabilizing p53 by inhibiting p53 ubiquitination and degradation (Fig. 7). Our work thus provides mechanistic insights into EGCG-induced p53-dependent cell apoptosis and cancer prevention. The interaction between EGCG and p53–NTD is highly dynamic and involves multiple binding interfaces. This adds to growing examples of dynamic interactions between IDP and small molecules[54,56–58]. Our data suggest that the NTD of p53 may be a target for cancer drug discovery.

## Methods

**Materials**. EGCG (458.4 Da, purity ≥95%) from green tea was ordered from Sigma-Aldrich (St. Louis, MO). Full-length p53, p53 NTD, full-length MDM2 and

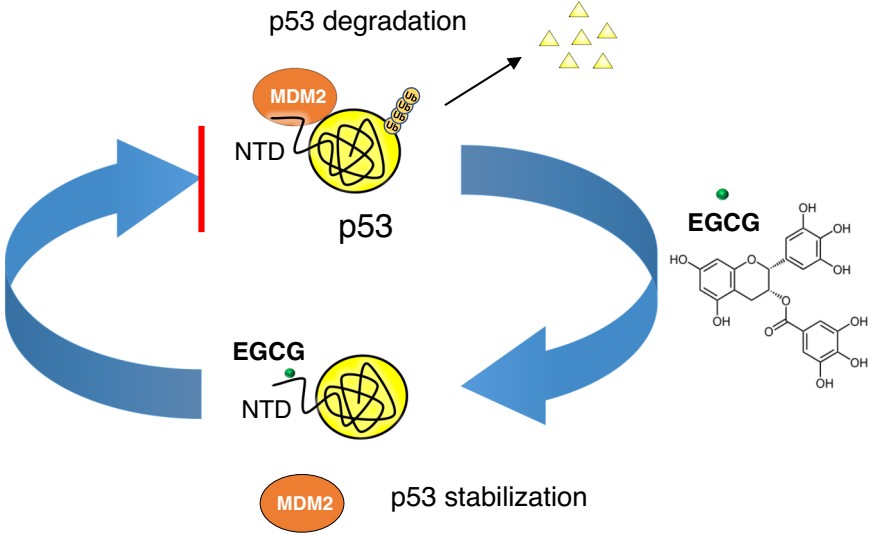

**Fig. 7 EGCG disrupts p53–MDM2 interaction by binding to the N-terminus of p53.** The EGCG–p53 interaction disrupts p53 interaction with its regulatory E3 ligase MDM2 and inhibits the ubiquitination of p53 by MDM2, likely stabilizing p53 for anti-tumor activity.

MDM2 p53 binding domain were overexpressed and purified[59]. Briefly, p53 NTD coding sequences were inserted downstream of the *Thermoanaerobactor tencongenesis* ribose-binding protein (RBP) gene, fused with a nucleotide sequence that harbors an HRV3C protease cleavage site. A His-tag sequence was placed at the 3′ end of the NTD gene to facilitate purification of the full-length fusion protein. The fusion gene was placed in the pET41 plasmid (Novagen) and transformed into *E. coli* BL21(DE3). Proteins were expressed at 18 °C with induction by IPTG and purified on a Ni$^{2+}$-NTA column. Following overnight cleavage by HRV3C protease (4 °C), the NTD proteins were recovered by Ni$^{2+}$-NTA and S75 size-exclusion chromatography to a final purity of >95% as judged by SDS–PAGE. The NTD constructs contain GPG and LE(H)$_8$ sequences at the N- and C-termini, respectively. SPR CM5 sensor chip and amine coupling kits were obtained from GE Healthcare Bio-Sciences AB (Uppsala, Sweden). Deuterium oxide (D, 99.9%) was purchased from Cambridge Isotope Laboratories, Inc. (Tewksbury, MA).

**EGCG–p53 interaction by SPR**. The binding behavior of EGCG with p53 was measured using SPR on a BIAcore 3000 system. First, full-length p53 or NTD of p53 was immobilized on a CM5 chip through its amine groups using EDC/NHS according to the standard amine coupling protocol. A 10 μL solution of 0.1 mg/mL full-length p53 (or NTD) was injected over the flow cell at 5 μL /min. Successful immobilization of p53 was confirmed by an ∼5000 resonance unit (RU) increase in the sensor chip. The first flow cell (control) was prepared without injection of full-length p53 (or NTD). After immobilization, the EGCG sample was diluted in running buffer (20 mM Tris–HCl, 150 mM NaCl, 1 mM TCEP, pH 7.2). Different dilutions of EGCG were injected at a flow rate of 30 μL/min for 3 min. Following sample injection, running buffer was passed over the sensor surface for a 3-min period for dissociation. The sensor surface was regenerated by a 20 μL injection of 10 mM glycine–HCl solution (pH = 2.25). The response was determined as a function of time (sensorgram) at 25 °C. Data were analyzed using Bioevaluation software, version 4.1.1 (GE Healthcare).

**Binding epitopes mapping by STD NMR**. STD spectra were recorded at 25 °C on a Bruker 600 MHz NMR spectrometer. NMR data were processed and analyzed using Topspin 3.5pl7, Topspin 4.0.6, and Mnova 12.0.3. 1 mM EGCG was mixed separately with 5 μM full-length p53 and 10 μM p53 NTD, dissolved in 20 mM Tris–HCl, 150 mM NaCl, 1 mM TCEP at pH 7.2 in 90/10% H$_2$O/D$_2$O. 1D STD-NMR spectra were recorded with 256 scans and selective saturation of protein resonances at 1 ppm using a series of Gaussian-shaped pulses, for a total saturation time of 2.0 s. A 30 ms spin lock was also employed to suppress p53 signals that overlap EGCG resonances. Saturation transfer reference (STR) spectrum was recorded with off-resonance frequency set to −10 ppm. STD spectrum of 1 mM EGCG alone without any protein was conducted under the same parameters as a negative control. STD amplification factors (STD$_{af}$) of each EGCG proton peak was calculated as $STD_{af} = \frac{I_{STD}}{I_0} \times \frac{[EGCG]_T}{[P]}$, where $I_{STD}$ represents the signal intensity in the STD spectrum, $I_0$ is the peak intensity in the STR spectrum, $[EGCG]_T$ is the total EGCG concentration and $[P]$ is the total protein concentration. Normalized STD$_{af}$ was calculated for each involved proton with the highest intensity signal set to 100%[33].

**NMR titration**. 2D NMR spectra of p53 NTD were acquired at 25 °C on a Bruker 800 MHz NMR spectrometer equipped with a cryogenic probe. NMR data were processed and analyzed using Topspin 3.5pl7 and Sparky 3.115. $^{15}$N-labeled or $^{15}$N,$^{13}$C-doubly labeled p53 NTD was dissolved in 25 mM NaCl, 50 mM Na$_2$HPO$_4$, 1 mM EDTA and 2 mM DTT at pH 6.8 in 90/10% H$_2$O/D$_2$O. A series of 2D $^{15}$N–$^{13}$C NCO spectra were performed on a 0.3 mM p53 NTD sample by adding increasing amounts of EGCG. 3D HNCACB, HNCO, (H)CANCO, and (H) CANCO-I experiments were recorded for the assignment of $^{15}$N–$^{13}$C NCO spectrum. Chemical shift perturbation (CSP) for amide nitrogen (N$^H$) and carbonyl carbon (C') chemical shifts upon EGCG binding were calculated by the difference between chemical shifts of the free and bound form of p53 NTD, represented by $\Delta\delta N^H$ and $\Delta\delta C'$, respectively.

**MD simulation**

*Simulation protocols.* p53 NTD (residues 1–61) were simulated with the N- and C-termini of the peptide capped with an CH$_3$CO-group and CH$_3$NH-group, respectively. Note that the construct was slightly shorter than the one used in experimental studies, since simulating such a construct allowed us to reduce computational cost. The latest a99SB-disp force field[45] was used to model peptide, water, and ions. This force field has been shown to be highly accurate in describing both local and long-range structural properties of p53 NTD[46]. To study the interactions between p53 NTD and EGCG, another simulation was performed, with one p53 NTD and one EGCG molecule placed in a simulation box. The EGCG molecule was modeled with the general AMBER force field[60]. For both NTD and NTD/EGCG systems, we have performed simulations using an advanced sampling technique, REST2[43,44], which allowed for accelerated simulation convergence while significantly reduced computational cost compared with temperature replica exchange method. The starting structures of NTD simulation were fully extended, while those of NTD/EGCG simulations were representative NTD structures identified from previous studies[46], with EGCG molecule randomly placed in the simulation box. The starting structure was solvated using about 23,500 water molecules in a truncated octahedron box, with a volume of ∼710 nm$^3$. This resulted in an effective concentration of 2.3 mM NTD (and EGCG in NTD/EGCG simulations). 14 Na$^+$ ions were added to each system as the counter ions to balance the charge of NTD.

All simulations were carried out using GROMACS 2019[61,62] patched with PLUMED 2.52[63–65]. Energy minimization using the steepest descent algorithm was first conducted to remove potential steric clashes in the initial conformation. Each system was then equilibrated at 298 K under NVT conditions for 100 ps, with the position of protein heavy atoms restrained by harmonic potentials with a force constant of 1000 kJ/mol/nm$^2$ in $x$, $y$, and $z$ directions. Another 1 ns NPT simulation was followed at 298 K and 1 atm, keeping the position of protein heavy atoms restrained. During the last stage of equilibration, the system was simulated under the same NPT condition for 1 ns, allowing all components of the system to freely move. Mean volume of the system was calculated for this simulation, and the conformation whose volume was closest to the mean volume was selected as the starting conformation for later REST2 simulations. All REST2 simulations were performed under NVT condition at 298 K, and only peptide (and EGCG, if present) was subject to effective tempering. This was achieved by scaling solute–solute and solute–solvent interactions by $\lambda$ and $\sqrt{\lambda}$, respectively. Each

REST2 simulation was comprised of 16 replicas with their $\lambda$ values ranging from 1.0 to 0.6, which corresponded to effective solute temperatures spaced exponentially between 298 and 500 K. Exchange between neighboring replicas was attempted every 2 ps, and the averaged exchange acceptance ratio for NTD and NTD/EGCG simulations were 28% and 12%, respectively. In all simulations, short-range nonbonded interactions were truncated at 1.2 nm, and long-range dispersion corrections were applied to van der Waals interactions. The particle mesh Ewald (PME) method[66] was used to treat long-range electrostatic interactions. The LINCS algorithm[67] was applied to constrain lengths of all bonds involving hydrogen atoms, which allowed us to integrate the equation of motion with 2 fs time step. Total simulation times for NTD and NTD/EGCG simulations were 3.4 and 3.9 μs/replica, making it one of the longest explicit solvent simulations of p53–NTD.

**Analysis**. The first 500 ns trajectories of each REST2 simulation were excluded from all analyses since they corresponded to the initial equilibration stage. Conformations sampled at condition of $\lambda = 1$ were collected to construct the structural ensemble of interest. All structural analyses were performed using GROMACS tools[61,62] and in-house scripts unless otherwise specified. To visualize potential conformational changes of p53 NTD induced by EGCG binding, we have performed principal component analysis (PCA) using peptide backbone heavy atoms. Featurization was first performed according to DRID, distribution of reciprocal of interatomic distances[68], as implemented in MSMBuilder[69]. PCA was then carried out on the combined trajectories of NTD and NTD/EGCG simulations. To examine possible binding modes of EGCG to NTD, another PCA analysis was performed on NTD/EGCG trajectory. For this, heavy-atom distances between NTD and EGCG were first computed. These distances were then transformed using a sigmoid function: $c = -1/(1 + \exp(-3*(r - 1.5))) + 1$, where $r$ is distance in Å, and $c$ indicates effectiveness of contact ranging from 0 to 1. After such featurization, PCA analysis was performed, which revealed various interaction patterns of NTD/EGCG complex regardless of the NTD conformations.

*SAXS experiments*. NTD construct 20–70 was purified as described above. The monodispersity of protein was confirmed by a single peak in DLS, as shown in the Supplementary Fig. 15.

Purified NTD was dialyzed against the following buffer: 50 mM NaCl, 25 mM K$_2$HPO$_4$, 1 mM EDTA, 2 mM DTT, pH 7 + 1% glycerol. The dialysis buffer was used as buffer match for SAXS experiments as well as to dissolve EGCG which was added to NTD. Therefore, all the SAXS samples, with and without EGCG, were in the exact same buffer. The protein concentration of the NTD in all SAXS samples was 580 μM, and samples of the complex contained NTD:EGCG in 1:1 or 1:2 molar ratio.

Prior to data collection, samples were thawed, filtered (pore size 0.2 μm) and centrifuged (30 min, 21,700 × g, 4 °C). SAXS data was collected at the beam line 7A1 at the Cornell high energy synchrotron source (CHESS), with a MacCHESS EIGER 4M detector (Dectris), at a single detector position, on February 10, 2020. Quartz capillary with a path length of 1.48 mm was used as the sample cell (OD = 1.5 mm, wall thickness = 10 μm). For each dataset, 20 frames were collected at 4 °C, with 0.5 s exposure times and 10-fold attenuation of the beam (wavelength = 9.9281 keV, beam dimensions = 250 × 250 μm, beam current = 50.3 mA, beam flux = 3 × 10$^{11}$). Most samples showed no detectable radiation damage, which was monitored by averaging 20 frames.

SAXS data were processed with the BioXTAS RAW software suite (versions 1.6.3 and 1.6.4)[70]. To obtain scattering intensity profiles, 20 data frames were reduced to scattering intensity profiles, placed on an absolute scale, averaged, and the scattering intensity profile of the buffer match was subtracted. The data quality was assessed by Guinier plots and dimensionless Kratky plots in BioXTAS RAW and molar mass calculations in PRIMUS[70–74] using ATSAS 3.0.0-3 software suite[76] (implemented in RAW[70]). Pair distance distribution $p(r)$ functions were derived from the scattering intensity profiles by the program GNOM[75] of the ATSAS 3.0.0-3 software suite[76] (implemented in RAW[70]). Fifteen bead model 3D reconstructions were performed with the Dammiff program[77], (implemented in ATSAS/RAW[70,76]. The resulting models were aligned, grouped into clusters, averaged, and the average model was refined in Dammiff[77–79]. Figures of the refined molecular envelopes were created in the program UCSF Chimera (version 1.9)[80], developed by the Resource for Biocomputing, Visualization, and Informatics at the University of California, San Francisco, with support from NIH P41-GM103311. For Fig. 3, the $p(r)$ functions were normalized to the highest signal of the NTD/EGCG 1:2 plot.

*Sedimentation velocity-analytical ultracentrifugation*. P53-NTD (20–70) in the presence and absence of EGCG were loaded into 3 or 12-mm two-sector charcoal-filled Epon centerpieces with Safire windows. All experiments were carried out at 20 °C using a Beckman–Coulter Proteomelab XL-A analytical ultracentrifuge equipped with absorbance optics and a 4-hole An-60 Ti rotor at 290,000 × g that was preequilibrated at 20 °C. The samples were scanned with the time interval between scans set to zero. Lamm equation modeling of all SV-AUC results was performed using the continuous distribution $(c(s))$ method in SEDFIT[40]. ME regularization using a confidence level of $P = 0.68$ was performed to identify the most parsimonious distribution consistent with the data, and the fits for each experiment gave acceptable RMSD values ranging between 0.003 and 0.01 (Supplementary Fig. 6). Three concentrations of p53–NTD ranging from 30–300 μM

were run and the resulting $c(s)$ plots showed that the sample was monodisperse with no evidence of oligomerization. Density, viscosity, and partial specific volume values, as well as frictional coefficients, were estimated using SEDNTERP[42].

To determine the impact of EGCG on the hydrodynamic properties of p53–NTD, 30 μM p53–NTD in the presence of 30 or 300 μM EGCG were run, and the data was analyzed using the Bayesian[41] implementation in SEDFIT using the s-values of EGCG (0.2) and p53–NTD (0.89) as prior expectations. Prior expectations were implemented as Gaussians in SEDFIT with a peak width of sigma = 0.2s and centered at the weight-average s-value of the main peak observed in the individual experiments with an amplitude of 0.05 OD units. All SV-AUC plots were created with GUSSI[81].

*Competition SPR*. Inhibitory effect of EGCG on p53–MDM2 interaction was detected by a solution/surface competition experiment performed on SPR, a well-established method which has been used in many studies for characterizing protein/ligand interaction[82,83]. The p53 binding domain of MDM2 was immobilized on a CM5 chip by amine coupling. First, SPR was used to study the kinetics and binding affinity of the interaction between MDM2 and full-length p53 as described before. Competition SPR was then performed using the MDM2 immobilized chip with p53 pre-incubated with EGCG in solution. Once the EGCG-binding sites on p53 are occupied by EGCG in the solution, p53 binding to the surface-immobilized MDM2 will decrease, resulting in a reduction in the magnitude of the SPR signal. With the increase of concentration of EGCG in the solution, the binding signal to the chip will decrease. IC$_{50}$ was estimated by the EGCG concentration at which inhibition percentage is half of the maximum.

*Fluorescence polarization anisotropy*. Fluorescence anisotropy measurements were made in opaque, round-bottom 96-well polystyrene plates using a Molecular Devices i3x plate reader with the Fluorescein-FP cartridge. N-terminally fluorescein-labeled MDM2 was prepared by treating MDM2 with 1 equivalent of NHS-fluorescein (Thermo Fisher Scientific) in 20 mM tris-buffered 150 mM saline + 1 mM TCEP (TBST) at pH 7.2 and incubating overnight at 4 °C before desalting (DoL ~ 0.5). Experiments were conducted using 10 nM fluor–MDM2 and performed in TBST at pH 7.2 plus the indicated concentration. Full-length p53 was desalted into the same buffer, and two-fold serial-dilutions were prepared in the 96-well plate. Experimental measurements of parallel and perpendicular fluorescence were blank subtracted using a pair of matched dilution series prepared without fluor–MDM2 to correct for background fluorescence, then converted to anisotropy. The results were plotted versus p53 concentration and fit to a one-site binding equation. $K_D$ and standard error values were determined by inverse-variance weighted pooling of the fit parameters from two replicates.

*Ubiquitination assay*. N-terminally fluorescein-labeled p53 was prepared by treating p53 with 1 equivalent of NHS-fluorescein (Thermo Fisher Scientific) in 20 mM tris-buffered 150 mM NaCl and 1 mM TCEP (TBST) at pH 7.2 and incubating overnight at 4 °C before desalting (DoL ~ 0.5). MDM2 and a 10:1:1 mixture of fluor–p53, E1 (hUBA1), and E2 (UbcH5c) were dialyzed against low-salt (50 mM NaCl) TBST. Lyophilized ubiquitin was directly resuspended in the same buffer. Reaction mixtures were prepared on ice, containing 2 μM MDM2, 0.5 μM fluor–p53, 200 μM ubiquitin, 5 mM MgATP, and varying concentrations of EGCG. The reaction mixtures were incubated at 20 or 37 °C for 4 or 2 h, respectively, then quenched by the addition of 4× SDS–PAGE-loading buffer and 25 mM TCEP. These samples were heated to 60 °C before being separated on a 4–20% gradient SDS–PAGE gel at 250 V for 40 min. The gels were imaged on an Azure Biosystems Sapphire biomolecular imager. The fluorescein channel was exported, and the extent of ubiquitination was quantified in BioRad ImageLab 6.1. The fluorescence fractions of the different Ub$_n$-p53 ($n = 0, 1, 2, >2$) species were plotted versus experimental EGCG concentration and fit to a four-parameter sigmoid to determine the apparent IC$_{50}$.

**Reporting summary**. Further information on research design is available in the Nature Research Reporting Summary linked to this article.

## Data availability
Data supporting the findings of this manuscript are available from the corresponding author upon reasonable request. A reporting summary for this Article is available as a Supplementary Information file. Source data are provided with this paper.

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

## Acknowledgements

The authors would like to acknowledge support from NIH R01CA206592 to C.W., NIH-NIA Training Grant (T32AG057464 to L.G.), NIH (R35-GM127040 to Y.Z., R01CA140522 (to M.S.C.) and 1R15GM128119-01 to S.R.S.). SAXS data was collected at beamline 7A1, Cornell High Energy Synchrotron source, supported by NSF award DMR-1829070, and by NIH/NIGMS award GM-124166. We thank Qingqiu Huang and Richard Gillilan for user support at the synchrotron source.

## Author contributions

C.W. and J.Z. designed the research. J.Z. carried out SPR and NMR experiments and data analysis. A.B. carried out fluorescence anisotropy and ubiquitination assays. Xia.L. and J.C. carried out MD simulation and data analysis. J.H. designed p53 NTD and MDM2 constructs. A.J.C., M.C. and M.S.C. carried out AUC experiments. S.R.S. carried out SAXS assay and data analysis. L.G. carried out gel filtration and DLS assays. J.Z., A.B., and Y.X. carried out the purification of p53 constructs. Xin.L. helped with NMR spectra assignment of NTD. W.J. and L.Y. helped with SPR control experiments. C.Y. and Y.Z. assisted with MD simulation. D.B. assisted with NMR data interpretation. C.W. and J.Z. wrote the paper. C.W., J.C., and S.N.L supervised the project.

## Competing interests

The authors declare no competing interests.
