## [Peer Review File · Nature Communications]

Reviewer #1 (Remarks to the Author):

The article "EGCG Binds Intrinsically Disordered N-terminal Domain of p53 and Disrupts p53-MDM2 Interaction" by Zhao et al describes an interesting study of the binding of EGCG to P53.

EGCG has attracted considerable interest in the biochemical community for a number of reasons, including the prevention of protein aggregation into amyloids, which is relevant for neurodegenerative diseases.

In this work, the authors perform a comprehensive biophysical-structural analysis to study the mechanism by which EGCG suppresses the binding between p53 and MDM2, which is a relevant interaction the context of cancer. The study identifies a major binding site of EGCG in the NTD region of p53 and provides a model of the inhibition of the p53-MDM2 interaction.

I believe the findings of this study are important for the research on p53, which is a relevant system in cancer and cellular signalling. In this context, the research will have a significant impact. One aspect that should be revised is the discussion of the results in terms of general IDP binding by EGCG or other small molecules, as general binding properties of EGCG to disordered proteins were already known in literature from studies of proteins such as alpha-synuclein or a-beta. Thus I suggest the focus of the paper should be specifically on the p53 interaction, which is the innovative part of the research.

I found some technical points that should be addressed by the authors:

Major points

1) The simulations show an increase in alpha-helical structure of the NTD upon EGCG interaction. This increase is substantial, with some regions reaching ~50% of the helical population in the presence of EGCG. Changes in chemical shifts (CS) upon EGCG binding are however modest, with a peak of 0.2 ppm in the C' for residue W23, where the simulations show the maximum helical increase. These CS changes do not justify such an increase in alpha-helix, as shown in the MD. It is therefore likely that biases from the MD force field have overpopulated the helical state. I have two suggestions for improvements.

i) the authors should analyse the secondary structure using CS. There are methods (e.g. delta2D) that provide populations of secondary structure elements from CS. This will allow to compare the helical propensity in the MDs with the helical content probed by CS.

ii) In case a force field bias is found in point (i), another force field should be used to check consistency of the MD results, including the comparison with NMR data.

2) Previous NMR studies of EGCG binding to IDPs have shown some resonance broadening effects. This aspect is not discussed in the paper, and could be an added value to the characterisation of the binding mechanism. It could also be interesting to measure transverse relaxation in the presence and absence of EGCG to be more quantitative.

3) The discussion of the data as a general IDP targeting mechanism is overstated (e.g. "Our data also have important implications on targeting intrinsically disordered proteins (IDP) by small molecules with dynamic interactions"). The dynamic nature of the binding of EGCG was shown already in literature with other IDP systems. For example a previous study showed the binding of EGCG to the same protein can vary from stable/localised to diffused/dynamical depending on the folding status of the protein (doi.org/10.1002/bip.23117).

I would therefore suggest to rewrite these element of discussion in view of current literature.

Minor points

4) lines 108-109, please add errors to the KD values. This should be applied throughout the text (e.g. line 248)

5) "The first 500 ns trajectories of each REST2 simulation were excluded from all analyses". Please specify what are the criteria of convergence and why 500 ns in the specific case.

Reviewer #2 (Remarks to the Author):

The authors describe the application of a number of methods to investigate the binding of Epigallocatechin gallate (EGCG) to the intrinsically disordered N-terminal domain of p53 (NTD). Experimentally, the authors use SPR to show that the binding of EGCG is relatively unchanged between full length p53 and NTD, and follow this up by showing the majority of shifts in the ¹⁵N-¹³C NMR in the presence of EGCG are in the NTD region. STD NMR is used to try identify key structural features of EGCG which bind to NTD. The authors employ MD simulations to support the hypothesis of conformational changes upon EGCG binding, and use SAXS to experimentally validate conformational changes. Finally the authors use a competition SPR experiment to observe the impact EGCG has on MDM2 p53 binding. The results suggest that the anti-tumour effect previously observed for EGCG may be due to an interaction with p53, and possibly prevent the degradation of p53 by MDM2. The in vitro concentrations required for binding to NTD (KD = 2.5 micromolar) are in a range that suggest this possibility from a nutritional perspective.

Overall the results disclosed in this manuscript are interesting and novel, and the SPR and NMR results validate one another. I do not have experience with SAXS and have limited experience with MD simulations, and so am unable to comment confidently on their application in this setting. A review of the literature on EGCG does not readily highlight a biological target with greater affinity than that observed by the authors, and these results do add key findings to the activity of a widely investigated polyphenol.

Major issues:

1) The STD NMR results as presented do not support the claims made in the text or figures. The STD NMR data highlights aromatic signals as being those with the most energy transfer, suggesting they are the areas of EGCG closest to the protein. The differences highlighted in Figure 1E do not show obvious differences between 2'/6' and 4a, and given the S/N of the STD recorded, the error in this measurement is likely to be quite high. No statistical analysis of the difference between the highlighted signals is described.

Additionally, the authors state that "hydrophobic interactions mediated by aromatic rings play an important role in EGCG-p53" based on the STD results. From looking at the described methodology, this technique is unlikely to show phenol resonances, nor the transfer difference to phenol residues. The results suggest to me that EGCG makes contacts across the entire structure. The STD results suggest very little about the orientation of EGCG binding, other than that EGCG does obviously interact with NTD.

I think the authors should comment on the errors, and significance of results associated with figure 1E, as well as the ability to observe phenols using this methodology, and adjust the claims made accordingly.

2) There is no validating assay for the inhibition of MDM2 binding to p53, nor a control. The competition SPR results show that EGCG prevents the binding of full length p53. In an ideal setting a second assay would be employed to validate these results (perhaps an ELISA assay, employing an antibody for p53). At a minimum a control should have been used to internally validate this assay. A known inhibitor of the MDM2-p53 interaction should also be put through the assay to observe what a positive result should look like, and a compound which is known to have no effect on the interaction should be used to ensure that the observed results are not due to some other effect.

Minor issues:

Some comments should be made in the introduction which help to put the binding constants observed here into context. What other reports are there of binding to cancer related targets, and at what binding constants have those studies observed?

Figure S6 is title "minimal chemical shifts". What is the criteria for "minimal"? How does that compare to the shifts observed for EGCG to p53. A table of shifts should be included for comparison of EGCG-MDM2 and EGCG-p53

line 290 should read "p53-NTD is highly dynamic" (currently high dynamic)

line 355 - p53 NTD is written twice

line 386 - u is used instead of the greek mu for micromolar

The manuscript is otherwise well presented, and will be of interest to a wide community. The modifications suggested above are required to provide confidence in the results described.

I look forward to reviewing the edited manuscript.

Regards,

Andrew Beekman

Reviewer #3 (Remarks to the Author):

A technical review of SAXS results in “EGCG Binds Intrinsically Disordered N-terminal Domain of p53 and Disrupts p53-MDM2 Interaction” by Zhao, et al.

Central to this study is the use of all-atom MD and SAXS to structurally characterize the intrinsically disordered N-terminal domain (NTD) of the p53 protein with and without EGCG. The authors briefly describe qualitative agreement between MD-derived P(r) curves and experimental P(r) from SAXS. However, no direct comparison is made, and if the authors wish to validate their MD with SAXS data there is significant discrepancy that must be resolved. There are additional points in the SAXS data analysis that need to be addressed. Specific comments and suggestions on how to address these issues are below:

- 1. How are the populated regions of the PCA plot sampled in the MD trajectory? If colored by time, are all of the structures in the space sampled evenly?**
- 2. Are the MD-derived P(r) curves normalized? Why do the integrated areas appear to be significantly different for NTD and NTD/EGCG? The shape of the P(r) for NTD/EGCG looks like one we would expect for an elongated protein, whereas that of the NTD alone is rather broad.**
- 3. The authors should validate the MD with SAXS before interpreting the MD results. In other words, the paragraph on SAXS should come before second MD paragraph. SAXS results should begin with description of analyses done (e.g. R_g and MW estimation) to assure and demonstrate that the samples likely represent monomers before going into the P(r) discussion.**
- 4. When first discussing the experimental P(r) curves, the shapes should be discussed as they contain surprisingly strong features for a flexible protein. Peaks at various length scales are mentioned but they are not correlated directly to structural features.**
- 5. Given the above points, Fig. 3B should be compared with SAXS more rigorously. The authors mention “qualitative agreement” of experimental P(r) functions with the MD, but the MD-derived P(r) of the NTD in particular shows poor agreement with experiment. Comparisons of a mixture of the shown MD P(r) functions with experimental results are also not shown, so it is difficult to discern agreement between the presented MD and experiment.**
 - a. What is the average hydrated R_g for the MD trajectories, and how do they compare to experiment?**
 - b. Does the R_g change throughout the trajectory? If so, is there a part that is in better agreement with experiment?**
 - c. Comparison of the experimental P(r) with those of the MD subsets shown in Fig. 3D-E may be informative in interpreting the features seen in SAXS P(r) curves and in identifying the source of the discrepancy between MD and SAXS.**
- 6. Conclusions about conformational compaction should not solely be drawn from bead model reconstructions. It is well known that dammif does a poor job of reconstructing extended or flexible structures. A more convincing argument should include comparison of features in Kratky representations of the different SAXS datasets as well as R_g/D_{max} values. The caveat is that changes to R_g/D_{max} made not be as informative as Kratky for this system since the protein is quite small.**
- 7. For the 1:1 mixture (as shown in figure S4), the residuals in the Guinier plot (Fig. S4B) have a visible trend, especially at the higher end of the q-range. Selecting a smaller range (for both low and high q-limits) to fit may be more appropriate.**

Comments outside of the SAXS/MD section:

- 1. NCO should be explicitly defined.**
- 2. How should the general reader evaluate the goodness of fit of your models to the SPR data?**

Specific comments on figures

1. Fig. 2B/C: Annotating with TAD1, TAD2, and PRD regions labeled would help with clarity.
2. Fig. 3A: W23 is hard to see.
3. Figs. 4, S4, S5: The authors should elaborate in the captions instead of referring simply to the text (likewise with figures S4 and S5).
4. Fig. 4A: Add Mw units and explain what the numbers mean (are they estimates from SAXS, and if so via which method?)
5. Fig. 4B: This should be removed from the main text, and preferably replaced with a more rigorous comparison as described above.

Reviewer #1

The article "EGCG Binds Intrinsically Disordered N-terminal Domain of p53 and Disrupts p53-MDM2 Interaction" by Zhao et al describes an interesting study of the binding of EGCG to P53.

EGCG has attracted considerable interest in the biochemical community for a number of reasons, including the prevention of protein aggregation into amyloids, which is relevant for neurodegenerative diseases.

In this work, the authors perform a comprehensive biophysical-structural analysis to study the mechanism by which EGCG suppresses the binding between p53 and MDM2, which is a relevant interaction in the context of cancer. The study identifies a major binding site of EGCG in the NTD region of p53 and provides a model of the inhibition of the p53-MDM2 interaction.

I believe the findings of this study are important for the research on p53, which is a relevant system in cancer and cellular signaling. In this context, the research will have a significant impact. One aspect that should be revised is the discussion of the results in terms of general IDP binding by EGCG or other small molecules, as general binding properties of EGCG to disordered proteins were already known in literature from studies of proteins such as alpha-synuclein or A-beta. Thus I suggest the focus of the paper should be specifically on the p53 interaction, which is the innovative part of the research.

I found some technical points that should be addressed by the authors:

Major points

1) The simulations show an increase in alpha-helical structure of the NTD upon EGCG interaction. This increase is substantial, with some regions reaching ~50% of the helical population in the presence of EGCG. Changes in chemical shifts (CS) upon EGCG binding are however modest, with a peak of 0.2 ppm in the C' for residue W23, where the simulations show the maximum helical increase. These CS changes do not justify such an increase in alpha-helix, as shown in the MD. It is therefore likely that biases from the MD force field have overpopulated the helical state. I have two suggestions for improvements.

- i) the authors should analyze the secondary structure using CS. There are methods (e.g. delta2D) that provide populations of secondary structure elements from CS. This will allow to compare the helical propensity in the MDs with the helical content probed by CS.
- ii) In case a force field bias is found in point (i), another force field should be used to check consistency of the MD results, including the comparison with NMR data.

We thank the reviewer for the very astute comments on the quantitative discrepancy of small CSP caused by EGCG and large increase in helicity seen in MD simulation.

We follow the reviewer's suggestion and estimated the helical using Delta2D. In apo NTD, Delta2D showed helical distributions similar to those revealed by MD (Fig. S9). Delta2D also showed an increase of up to 1% in helicity in residues ~50-55 upon EGCG binding.

Although both MD and CSP revealed helical increase in EGCG binding region around residue W53, MD has apparently over-estimated the helicity increase in NTD upon EGCG binding. This is mainly due to much greater challenge in sampling various bound conformations than the NTD apo state. The bound states have longer life-times and will persist once detected/formed during the simulation, leading to over-estimated population of the compact states with EGCG bound. More limited sampling of various bound states is also reflected in much greater uncertainty in the average residue helicity calculated for the bound state (see Fig. S9A). We have therefore moved the figures associated with increased helical content upon EGCG binding to SI, making it a minor point of the paper.

We greatly appreciate the reviewer's recommendation of testing/evaluating alternative force fields. In fact, we had previously tested several of the latest protein force fields for the simulation of NTD and reported our findings in a recent paper [Liu, Xiaorong, and Jianhan Chen. "Residual structures and transient long-range interactions of p53 transactivation domain: Assessment of explicit solvent protein force fields." *Journal of chemical theory and computation* 15.8 (2019): 4708-4720]. The accuracy of six latest force fields with explicit solvent in was compared in modeling p53-TAD. The a99SB-disp force field used in this study was found to generate disordered ensembles that have superior agreement with a wide range of experimental data, such as chemical shifts and PRE, smFRET and SAXS. As such, we chose to use a99SB-disp in this work.

2) Previous NMR studies of EGCG binding to IDPs have shown some resonance broadening effects. This aspect is not discussed in the paper, and could be an added value to the characterization of the binding mechanism. It could also be interesting to measure transverse relaxation in the presence and absence of EGCG to be more quantitative.

We now include ^{15}N relaxation experiments (R_1 , R_2 and NOE) on NTD in the presence and absence of EGCG. A significant increase in transverse relaxation rate R_2 value was observed in the presence of EGCG, without large changes in R_1 and NOE. R_2 ($1/T_2$) is directly correlated with the linewidth of the spectra ($\nu_{1/2} = R_2/\pi$), indicating the resonance broadening effects of EGCG on NTD. The increased R_2 and resonance broadening effect of EGCG could be due to the chemical exchange between different EGCG-bound conformation of NTD, consistent with the multiple bound conformations observed in MD simulation of EGCG-bound state.

We have added these data in SI and add the following sentences the main text:

"EGCG binding caused large increase in ^{15}N transverse relaxation in NTD (Fig. S11), likely caused by chemical exchange between multiple EGCG-bound states. This is consistent with the multiple bound conformations observed in MD simulation of EGCG-bound state.

3) The discussion of the data as a general IDP targeting mechanism is overstated (e.g. "Our data also have important implications on targeting intrinsically disordered proteins (IDP) by small molecules with dynamic interactions"). The dynamic nature of the binding of EGCG was shown already in literature with other IDP systems. For example a previous study showed the binding of EGCG to the same protein can vary from stable/localised to diffused/dynamical depending on the folding status of the protein (doi.org/10.1002/bip.23117).

I would therefore suggest to rewrite these element of discussion in view of current literature.

We've changed the statement in the abstract to 'The dynamic nature of the EGCG/NTD interaction offers further support to the notion that intrinsically disordered proteins (IDPs)—conventionally considered undruggable targets—might be effectively targeted using small molecules with dynamic interactions.' In the summary, we changed the statement to "The interaction between EGCG and p53-NTD is highly dynamic and involves multiple binding interfaces. This adds to growing examples of dynamic interactions between IDP and small molecules." and cited literatures [*Ban, D. et al. J. Am. Chem. Soc., 2017*][*Liang, C. et al. J. Chem. Theory Comput, 2019*][*Neira, J.L. et al. Sci. Rep, 2017*].

Minor points

4) lines 108-109, please add errors to the KD values. This should be applied throughout the text (e.g. line 248)

We've added errors to the KD values.

5) "The first 500 ns trajectories of each REST2 simulation were excluded from all analyses". Please specify what are the criteria of convergence and why 500 ns in the specific case.

We examined key structural properties (mainly helicity and Rg) and make sure that they largely reach plateaus before including the trajectories in the analysis. E.g., the convergence in Rg after 500 ns was already presented in the Fig. S1 in the submitted MS (now Fig. S8):

Reviewer #2

The authors describe the application of a number of methods to investigate the binding of Epigallocatechin gallate (EGCG) to the intrinsically disordered N-terminal domain of p53 (NTD). Experimentally, the authors use SPR to show that the binding of EGCG is relatively unchanged between full length p53 and NTD, and follow this up by showing the majority of shifts in the ¹⁵N-¹³C NMR in the presence of EGCG are in the NTD region. STD NMR is used to try identify key structural features of EGCG which bind to NTD. The authors employ MD simulations to support the hypothesis of conformational changes upon EGCG binding, and use SAXS to experimentally validate conformational changes. Finally the authors use a competition SPR experiment to observe the impact EGCG has on MDM2 p53 binding. The results suggest that the anti-tumour effect previously observed for EGCG may be due to an interaction with p53, and possibly prevent the degradation of p53 by MDM2. The in vitro concentrations required for binding to NTD (K_D = 2.5 micromolar) are in a range that suggest this possibility from a nutritional perspective.

Overall the results disclosed in this manuscript are interesting and novel, and the SPR and NMR results validate one another. I do not have experience with SAXS and have limited experience with MD simulations, and so am unable to comment confidently on their application in this setting. A review of the literature on EGCG does not readily highlight a biological target with greater affinity than that observed by the authors, and these results do add key findings to the activity of a widely investigated polyphenol.

Major issues:

1) The STD NMR results as presented do not support the claims made in the text or figures. The STD NMR data highlights aromatic signals as being those with the most energy transfer, suggesting they are the areas of EGCG closest to the protein. The differences highlighted in Figure 1E do not show obvious differences between 2'/6' and 4a, and given the S/N of the STD recorded, the error in this measurement is likely to be quite high. No statistical analysis of the difference between the highlighted signals is described.

Additionally, the authors state that "hydrophobic interactions mediated by aromatic rings play an important role in EGCG-p53" based on the STD results. From looking at the described methodology, this technique is unlikely to show phenol resonances, nor the transfer difference to phenol residues. The results suggest to me that EGCG makes contacts across the entire structure. The STD results suggest very little about the orientation of EGCG binding, other than that EGCG does obviously interact with NTD. I think the authors should comment on the errors, and significance of results associated with figure 1E, as well as the ability to observe phenols using this methodology, and adjust the claims made accordingly.

Error bars have now been calculated from the S/N of NMR spectra and presented in Fig. 1E.

We agree with the reviewer's astute comments that STD is mainly a confirmatory assay for binding in the solution phase. Because phenol hydroxyl resonances cannot be detected in NMR due to fast solvent exchange, STD indeed does not provide much insight as to the binding orientation of EGCG. We have added these sentences associated with STD as follows:

"Protons with the largest STD_{ar} are distributed throughout the EGCG molecule, indicating that most parts of EGCG molecule is involved in NTD interaction. However, phenol hydroxyl resonances cannot be detected in solution NMR due to fast solvent exchange; thus the role of these hydroxyl group cannot be established based on STD data."

2) There is no validating assay for the inhibition of MDM2 binding to p53, nor a control. The competition SPR results show that EGCG prevents the binding of full length p53. In an ideal setting a second assay would be employed to validate these results (perhaps an ELISA assay, employing an antibody for p53). At a minimum a control should have been used to internally validate this assay. A known inhibitor of the MDM2-p53 interaction should also be put through the assay to observe what a positive result should look

like, and a compound which is known to have no effect on the interaction should be used to ensure that the observed results are not due to some other effect.

We thank the reviewer for suggesting these crucial experiments.

The competition SPR experiments we used here were well-established in our previous work. [Zhao, Jing, et al. "Glycan Determinants of Heparin-Tau Interaction." *Biophysical Journal* 112.5 (2017): 921-932.] [Zhao, Jing, et al. "Kinetic and Structural Studies of Interactions between Glycosaminoglycans and Langerin." *Biochemistry* 55.32 (2016): 4552-4559.]

A negative control experiment was carried out using Vitamin C, another popular dietary supplement, which didn't affect the binding of p53 to MDM2 at 0.125 μM , 0.5 μM and 1 μM . This is to ensure that the observed EGCG results are not due to some other effect.

For positive control, we used Nutlin-3a, a well-known peptidomimetic inhibitor for p53/MDM2 interaction. Due to the poor solubility of Nutlin-3a in water, SPR is challenging to perform. We have used fluorescence anisotropy assay to demonstrate the inhibition of full length p53/MDM2 interaction by EGCG, along with Nutlin-3a (see figure below). These positive and negative controls data were added to the supporting information.

For further validation of EGCG inhibition of MDM2 binding to p53, we have carried out *in vitro* p53 ubiquitination assay mediated by MDM2 and demonstrated the inhibitory effect of EGCG with IC_{50} of $\sim 100 \mu\text{M}$ (see figure below).

Minor issues:

Some comments should be made in the introduction which help to put the binding constants observed here into context. What other reports are there of binding to cancer related targets, and at what binding constants have those studies observed?

We've added comments on EGCG binding to cancer-related targets in the Introduction as below.

“At the molecular level, EGCG has been demonstrated to interact with cancer-related proteins, such as glucose-regulated protein 78 (GRP78) and Ras-GTPase-activating protein SH3 domain-binding protein 1 (G3BP1), with $\sim\mu\text{M}$ affinities.” [Ermakova, S. P. et al. (-)-Epigallocatechin gallate overcomes resistance to etoposide-induced cell death by targeting the molecular chaperone glucose-regulated protein 78. *Cancer Res.* 66, 9260–9269 (2006).] [Shim, J. et al. Epigallocatechin Gallate Suppresses Lung Cancer Cell Growth through Ras – GTPase-Activating Protein SH3 Domain-Binding Protein 1. *Cancer Prev Res.* 3, 670–680 (2010).]

Figure S6 is title "minimal chemical shifts". What is the criteria for "minimal"? How does that compare to the shifts observed for EGCG to p53. A table of shifts should be included for comparison of EGCG-MDM2 and EGCG-p53

We've added the spectra of NTD perturbed by EGCG side by side with MDM2, which showed clearly more changes on NTD induced by EGCG (Fig. S14A, see below). Much more significant chemical shift perturbations were induced by NTD than that of MDM2, in both N and H dimension, which were compared in (Fig. S14B, see below). We've modified the title by changing "minimal chemical shifts perturbations" to "CSPs comparison".

line 290 should read "p53-NTD is highly dynamic" (currently high dynamic)

Changed

line 355 - p53 NTD is written twice

Changed

line 386 - u is used instead of the greek mu for micromolar

Changed

The manuscript is otherwise well presented, and will be of interest to a wide community. The modifications suggested above are required to provide confidence in the results described.

Reviewer #3

A technical review of SAXS results in “EGCG Binds Intrinsically Disordered N-terminal Domain of p53 and Disrupts p53-MDM2 Interaction” by Zhao, et al.

Central to this study is the use of all-atom MD and SAXS to structurally characterize the intrinsically disordered N-terminal domain (NTD) of the p53 protein with and without EGCG. The authors briefly describe qualitative agreement between MD-derived $P(r)$ curves and experimental $P(r)$ from SAXS. However, no direct comparison is made, and if the authors wish to validate their MD with SAXS data there is significant discrepancy that must be resolved. There are additional points in the SAXS data analysis that need to be addressed. Specific comments and suggestions on how to address these issues are below:

1. How are the populated regions of the PCA plot sampled in the MD trajectory? If colored by time, are all of the structures in the space sampled evenly?

Simulation of complex IDPs like p53-NTD and their ligand binding requires so-called enhanced sampling techniques in order to adequately sample the broad conformational space. In this work, we used replica exchange with solute tempering (REST). The output of REST is *the thermodynamic ensemble* of the protein conformation, without time or kinetic information. The PCA analysis shown in Fig 4 is derived from all snapshots sampled by REST at 300 K (except the first 500 ns equilibration stage). The probability of the system visiting each point in the conformational space is shown using color gradient.

2. Are the MD-derived $P(r)$ curves normalized? Why do the integrated areas appear to be significantly different for NTD and NTD/EGCG? The shape of the $P(r)$ for NTD/EGCG looks like one we would expect for an elongated protein, whereas that of the NTD alone is rather broad.

MD-derived $P(r)$ was not normalized in the previously submitted MS, leading to unequal area under the curve. We have now normalized $P(r)$ from MD, to have the highest peak in the bound state to 1, as done for the experimental $P(r)$ curves. $P(r)$ of the unbound state is scaled using the same ratio to ensure that the two curves have the same area under the curve.

3. The authors should validate the MD with SAXS before interpreting the MD results. In other words, the paragraph on SAXS should come before second MD paragraph. SAXS results should begin with description of analyses done (e.g. R_g and MW estimation) to assure and demonstrate that the samples likely represent monomers before going into the $P(r)$ discussion.

We have modified the text on SAXS accordingly and placed before our description of MD results. We have in addition carried out AUC experiments to further demonstrate the conformational compaction upon EGCG binding. SAXS and AUC data are discussed as below:

Small-Angle X-ray Scattering (SAXS) experiments were carried out to study large-scale conformational changes in the NTD induced by EGCG. Here we used a shorter construct NTD20-70 which includes the major binding sites of EGCG. Results from dynamic light scattering and Guinier plots of the SAXS data suggest that the NTD is monodisperse, with a radius of gyration of 23.9 Å (Figures S1-S3, Table S1A, Methods). The molar mass derived from the SAXS data and AUC shows that the apo NTD is a monomer (MW=6.4 kDa; while the expected MW based on sequence is 6.9 kDa). In pair distance distribution functions, $P(r)$, derived from SAXS profiles (Figure 3A), the profile of apo NTD displayed two peaks at distances of ~6 and 18 Å and a small bump at ~30 Å, suggesting an extended and elongated conformation with subdomains. Addition of EGCG (1:1 and 1:2) resulted in an increase of the relative height of the peak at a distance of 18 Å and lead to the decrease of the small bump at 30 Å, indicating a shift of the conformational equilibria induced by EGCG binding that results in compaction of the NTD. Bead models of apo NTD and EGCG-NTD were reconstructed from the SAXS $P(r)$ functions and showed a small but significant compaction along the length of the molecular envelopes (Figure S4). The normalized Kratky plots of the NTD show that upon addition of EGCG the signal is reduced at high q values, further supporting that binding of EGCG to the NTD shifts the equilibrium from extended to more compact folding conformations (Figure 3B).

In addition, AUC was carried out to provide additional evidence for conformational compaction upon binding to EGCG. Continuous size distribution analysis ($C(s)$) by SEDFIT and Bayesian analyses were used to analyze the sedimentation velocity data of four samples: EGCG, NTD, EGCG+NTD at 1:1 molar ratio, and EGCG + NTD at 10:1 ratio (Fig. 3C and Fig. S5). As expected, EGCG has very small sedimentation coefficient (grey curve) as a small molecule; while apo NTD has a sharp peak centered around 0.8 S (red curve), clearly demonstrating that apo NTD is a monomer. With the addition of EGCG, a new peak at ~1.4 S began to appear (blue and black curve, inset), showing EGCG concentration dependent shifting of the p53 NTD. The frictional ratio of apo-P53-NTD is 1.5, consistent with an elongated or intrinsically disordered protein. However, the peak at 1.4 s gives a frictional ratio of ~1.1 for a one-to-one complex between P53-NTD and EGCG (with estimated MW of 7.4 kDa). This peak cannot be a NTD dimer, as a compact dimer with frictional ratio of 1.1 would sediment at 1.9 S. Thus, the smaller frictional ratio of the 1.4 S peak demonstrates that a subpopulation of the EGCG-bound NTD has a more compact conformation upon binding EGCG, in agreement with data from SAXS.

4. When first discussing the experimental $P(r)$ curves, the shapes should be discussed as they contain surprisingly strong features for a flexible protein. Peaks at various length scales are mentioned but they are not correlated directly to structural features.

Addressed above.

5. Given the above points, Fig. 3B should be compared with SAXS more rigorously. The authors mention “qualitative agreement” of experimental $P(r)$ functions with the MD, but the MD-derived $P(r)$ of the NTD in particular shows poor agreement with experiment. Comparisons of a mixture of the shown MD $P(r)$

functions with experimental results are also not shown, so it is difficult to discern agreement between the presented MD and experiment.

We have added a figure for directly comparing four $P(r)$ functions: two experimental (NTD (black) and NTD/EGCG (red), dotted curves) and two MD-derived curves (smooth curves). We emphasize that the agreement is only *qualitative* in the decrease of the bump near 30 Å and the increase of a peak near 20 Å.

a. What is the average hydrated R_g for the MD trajectories, and how do they compare to experiment?

Averaged R_g at 298 K is ~ 2.6 nm, which agreed very well with R_g fit from $P(r)$ presented in Supporting table 1 in the range of 24.9 to 26.5 Å.

b. Does the R_g change throughout the trajectory? If so, is there a part that is in better agreement with experiment?

As noted above, we used an enhanced sampling technique (REST) in order to sample the large and complex conformational space of p53-NTD. The range of R_g sampled is very broad. This is shown in Fig S7.

c. Comparison of the experimental $P(r)$ with those of the MD subsets shown in Fig. 3D-E may be informative in interpreting the features seen in SAXS $P(r)$ curves and in identifying the source of the discrepancy between MD and SAXS.

We greatly appreciate the reviewer's suggestion. We have presented the $P(r)$ calculated from highly populated conformations in the figure below and included them in SI. The double peak appearance of the experiment $P(r)$ can not be reproduced from a combination of three major conformations. We note that approaches have been proposed where *de novo* atomistic disordered ensembles are reweighted to better match SAXS (and other experimental observables). However, these approaches can be highly problematic due to the under-determined nature of such fitting for disordered proteins (in contrast to structured proteins with a small number of conformational states). These issues have been discussed in detail in the IDP literature (e.g., Chen, J., Archives of Biochemistry and Biophysics 2012; Fisher, C. K. and C. M. Stultz, Current Opinion in Structural Biology, 2011).

6. Conclusions about conformational compaction should not solely be drawn from bead model reconstructions. It is well known that dammif does a poor job of reconstructing extended or flexible structures. A more convincing argument should include comparison of features in Kratky representations of the different SAXS datasets as well as R_g/D_{max} values. The caveat is that changes to R_g/D_{max} made not be as informative as Kratky for this system since the protein is quite small.

We have added the Kratky plot, which showed clear reduction of high q values in EGCG bound sample, again demonstrating conformational compaction upon EGCG binding. In addition, we have used AUC to demonstrate conformational compaction upon EGCG binding. R_g and D_{max} values were already listed in Table S1A. As pointed by the reviewer, changes to R_g/D_{max} made are not informative for this system, likely because the protein is quite small.

7. For the 1:1 mixture (as shown in figure S4), the residuals in the Guinier plot (Fig. S4B) have a visible trend, especially at the higher end of the q -range. Selecting a smaller range (for both low and high q -limits) to fit may be more appropriate.

We've replaced figure S4 as the reviewer suggested.

Comments outside of the SAXS/MD section:

1. NCO should be explicitly defined.

We've defined NCO as 'amide nitrogen-carbonyl correlation' where it firstly appeared.

2. How should the general reader evaluate the goodness of fit of your models to the SPR data?

BIAevaluation provides a statistical parameter χ^2 , which is a measure of the average deviation of the experimental data from the fitted curve. Lower χ^2 values indicate a better fit. However, the χ^2 value strongly depends on the measured binding level and therefore a generally acceptable χ^2 cut-off cannot be established. χ^2 value less than 10% R_{max} (maximum binding response) has been accepted as a criteria to exclude bad fitting in literatures [Dicara, Danielle, et al. "High-throughput screening of antibody variants for chemical stability: identification of deamidation-resistant mutants." *mAbs* 10.7 (2018): 1073-1083.] [Wang, Limei, et al. "Novel interactomics approach identifies ABCA1 as direct target of evodiamine, which increases macrophage cholesterol efflux." *Scientific Reports* 8.1 (2018)]. We've added χ^2 values in the related figure captions, which were all within the acceptable range (10% R_{max}).

Specific comments on figures

1. Fig. 2B/C: Annotating with TAD1, TAD2, and PRD regions labeled would help with clarity.

Fig 2B/C was annotated with TAD1, TAD2, and PRD labelling.

2. Fig. 3A: W23 is hard to see.

We guess the reviewer means Fig. 2A. We've modified the W23 block to make it clearer.

3. Figs. 4, S4, S5: The authors should elaborate in the captions instead of referring simply to the text (likewise with figures S4 and S5).

Changed

4. Fig. 4A: Add Mw units and explain what the numbers mean (are they estimates from SAXS, and if so via which method?)

We've added Mw units (kDa) and explained the calculation methods in the figure caption.

5. Fig. 4B: This should be removed from the main text, and preferably replaced with a more rigorous comparison as described above.

We've moved beads model to the supporting information and replaced with Kratky plot.

Reviewer #1 (Remarks to the Author):

I believe the revised form of the manuscript "EGCG Binds Intrinsically Disordered N-terminal Domain of p53 and Disrupts p53-MDM2 Interaction" has addressed my main concerns.

The relaxation data add a fine characterisation for the EGCG binding and the analysis of d2D has shown some consistency with the MD data, despite the latter clearly overestimate the increase in helical population. As the authors have downplayed the description of the MD, due to the evident bias toward structured states, this is acceptable to be reported in the supplementary material with some levels of critical description.

Moreover, the remodulation of some sentences, previously overstating the conclusions from this study, has also improved the manuscript.

Reviewer #2 (Remarks to the Author):

The edited manuscript and addition of new data points has addressed all of my previous referee's comments and provided trust in the results presented.

I believe the work is suitable for publication as is.

Reviewer #3 (Remarks to the Author):

Technical review of "EGCG Binds Intrinsically Disordered N-terminal Domain of p53 and Disrupts p53-MDM2 Interaction".

In the abstract, the authors conclude the following finding:

“In addition, EGCG induces large-scale conformational change and overall compaction in NTD, which was confirmed by small angle X-ray scattering and analytical ultracentrifugation.”

Unfortunately, I don't see evidence for large-scale conformational changes in the SAXS data. Neither R_g nor D_{max} changes appreciably with addition of EGCG. Although changes are observed in the relative peak heights of the $P(r)$ curves, the newly added Kratky plots indicate that the construct is fully unfolded (consistent with an R_g estimate of ~ 22 Å for a 50-residue random coil). The Kratky plots more definitively indicate that there is no major compaction with addition of EGCG. The subtle changes that the authors cite in the high q region are within the experimental noise and difficult to justify as a clear case of compaction. As the Kratky plot indicates that the construct is unfolded, bead modeling is also not justified and should not be shown.

Concerningly, the AUC analysis suggests that the frictional ratio decreases from 1.5 (extended) to 1.1 (nearly perfectly spherical). I am not sure what to make of the discrepancy between AUC and SAXS (which indicates that the construct remains extended), except to say that the Fig. S5D does not show two

distinct sedimentation rates. Although in theory, the idea behind $c(s)$ is to deconvolute diffusion, if there are two distinct sedimentation rates, I would expect to see step-shaped concentration curves.

It is possible that EGCG can interact with the NTD without causing a significant change in conformation. The authors should consider what can be concluded strongly.

Responses to Reviewer #3's Comments

In the abstract, the authors conclude the following finding: “In addition, EGCG induces large-scale conformational change and overall compaction in NTD, which was confirmed by small angle X-ray scattering and analytical ultracentrifugation.”

Unfortunately, I don't see evidence for large-scale conformational changes in the SAXS data. Neither R_g nor D_{max} changes appreciably with addition of EGCG. Although changes are observed in the relative peak heights of the $P(r)$ curves, the newly added Kratky plots indicate that the construct is fully unfolded (consistent with an R_g estimate of ~ 22 Å for a 50-residue random coil). The Kratky plots more definitively indicate that there is no major compaction with addition of EGCG. The subtle changes that the authors cite in the high q region are within the experimental noise and difficult to justify as a clear case of compaction. As the Kratky plot indicates that the construct is unfolded, bead modeling is also not justified and should not be shown.

We would like to thank the reviewer 3 for the careful examination of the SAXS and AUC data.

We agree with the reviewer that SAXS data alone didn't show enough evidence for large-scale conformational changes. Both $P(r)$ and Kratky plot did show EGCG-dependent changes and a trend towards less disorder. Such changes seen in SAXS can be due to two possibilities: 1. EGCG induces a subtle conformational change in NTD; 2. EGCG induces emergence of a subpopulation of a compact conformation of NTD. SV-AUC, a technique with superior hydrodynamic resolution, was applied to resolve these two possibilities and Bayesian statistical analysis of AUC data supported the 2nd possibility. The compact conformation was also observed in MD simulation of EGCG-bound NTD (Fig. 4), although EGCG can also bind to disordered NTD. Combining the insights from all three techniques, SAXS, AUC and MD, we conclude that EGCG induces more conformational heterogeneity in NTD, with an emergence of a compact conformation of NTD. We have revised the manuscript to reflect these considerations and to clarify the rationale for the SV-AUC experiments.

We have changed the relevant sentence in the abstract to:

“In addition, SAXS showed that EGCG causes a trend towards less disorder in NTD, which is due to the emergence of a subpopulation of compact bound conformation by Bayesian analysis of AUC data. Consistently, MD simulation showed that EGCG brings about more conformational heterogeneity in NTD, where EGCG can bind to both disordered and compact conformations of NTD.”

We agree the bead models can be misleading and they have been removed from the SI.

Concerningly, the AUC analysis suggests that the frictional ratio decreases from 1.5 (extended) to 1.1 (nearly perfectly spherical). I am not sure what to make of the discrepancy between AUC and SAXS (which indicates that the construct remains extended), except to say that the Fig. S5D does not show two distinct sedimentation rates. Although in theory, the idea behind $c(s)$ is to deconvolute diffusion, if there are two distinct sedimentation rates, I would expect to see step-shaped concentration curves.

The SAXS data and AUC analysis are actually consistent with each other. The SAXS results are due to linear combination of disordered and compact conformation of the bound state of NTD. Both extended and nearly spherical conformations were observed in MD simulation of EGCG-bound NTD, as shown below:

We agree with the reviewer that a step-shaped curve would be expected if there were distinct conformational or oligomeric populations that interconvert slowly on the timescale of the AUC experiment ($k_{\text{off}} < 10^{-5} \text{ s}^{-1}$).

In examining Fig. S4 (previous Fig. S5), we realized that the panels were mislabeled and have been corrected in the new revision. We apologize for any confusion this has caused and thank the reviewer for catching the error, which has been corrected in the revised version. We note that the correct version of the 10:1 experiment shows evidence of multiple boundaries, as shown below:

However, the sedimentation profile of EGCG-NTD also shows evidence of interactions occurring on a more rapid timescale. Molecules interacting (or interconverting) under the rapid kinetic regime will not show a step-shaped curve but instead will show a single diffusionally broadened boundary that reflects sedimentation of bound and unbound species that cannot be resolved within the signal to noise of the experiment using the standard maximum entropy regularization routine in the typical $c(s)$ analyses. This is evident in the original $c(s)$ plots showing that EGCG induces broadening and shifting of the NTD sedimentation peak, as shown below. This plot has now been included in supplementary figures and described in the text.

This is the reason why we decided to use a Bayesian approach to fit the SV-AUC data. The Bayesian analysis uses prior knowledge about the sedimentation behavior of the macromolecules as constraints in the regularization to increase the sensitivity and resolution, such that the contributions of the different species can be resolved. This approach has been shown to be particularly useful for resolving boundaries composed of molecules interacting under a rapid kinetic regime (Brown, Balbo and Schuck, (2007) *Biomacromolecules* 8, 2011-24). We have relabeled the y-axis of Fig. 3C to $c^{(p)}(s)$ to reflect the use of prior knowledge in the calculations.

Lastly, we feel this approach provides a reasonable explanation for the perceived discrepancy between the SAXS and AUC experiments. The SV-AUC approach allowed us to characterize a wider stoichiometric range of EGCG/NTD ratios without the complications of a small population of aggregates dominating the signal, which precludes unambiguous analysis of SAXS data. The concentration dependence of the faster migrating species observed with the wider stoichiometric range bolsters the existence of a more compact form of NTD when bound with EGCG, which is supported by MD simulation as well. Such behavior would be not expected if the entire population of NTD molecules remained intrinsically disordered. While the SAXS experiments could only be conducted with up to a 2:1 ratio, the trends observed are fully consistent with the signal reflecting a linear combination of compact and disordered population of molecules.

It is possible that EGCG can interact with the NTD without causing a significant change in conformation. The authors should consider what can be concluded strongly.

We indeed agree that EGCG can bind to NTD in a disordered and extended conformation but EGCG binding also induces the emergence of a compact conformation of NTD, as shown by AUC and MD. As discussed above, we have clarified our methods and conclusions in the revised text.

Reviewer #3 (Remarks to the Author):

Technical review of "EGCG Binds Intrinsically Disordered N-terminal Domain of p53 and Disrupts p53-MDM2 Interaction".

The revised manuscript is significantly better. I have three final comments:

1. The authors state, "The normalized Kratky plots of the NTD show that upon addition of EGCG, the signal is modestly reduced at high q values in a concentration dependent manner (Figure 3B)." Subtle differences in the high-q region of a Kratky plot are very often due to background subtraction error. To clarify that the authors have excluded the possibility the trend is not due to such an issue, they should - at the very least - state how the buffers were prepared in the Methods for the samples with EGCG.
2. Related to the above, the Methods should be edited (to reflect the removal of the shape reconstructions).
3. Concentrations used in SAXS/AUC should be discussed in the main text and figure captions in the same units (μM).
4. Regarding AUC, I agree that peak broadening suggests that there is a fast exchange. If that is the case, could 1.4 S still be a lower bound for the sedimentation coefficient of the complex?

Technical review of "EGCG Binds Intrinsically Disordered N-terminal Domain of p53 and Disrupts p53-MDM2 Interaction".

The revised manuscript is significantly better. I have three final comments:

1. The authors state, "The normalized Kratky plots of the NTD show that upon addition of EGCG, the signal is modestly reduced at high q values in a concentration dependent manner (Figure 3B)." Subtle differences in the high- q region of a Kratky plot are very often due to background subtraction error. To clarify that the authors have excluded the possibility the trend is not due to such an issue, they should - at the very least - state how the buffers were prepared in the Methods for the samples with EGCG.
2. Related to the above, the Methods should be edited (to reflect the removal of the shape reconstructions).

Response: Thanks very much for raising this point. We've modified the Methods regarding SAXS sample preparation as follow:

Purified NTD was dialyzed against the following buffer: 50 mM NaCl, 25 mM K_2HPO_4 , 1 mM EDTA, 2 mM DTT, pH 7 + 1% glycerol. The dialysis buffer was used as buffer match for SAXS experiments as well as to dissolve EGCG which was added to NTD. Therefore, all the SAXS samples, with and without EGCG, were in the exact same buffer.

3. Concentrations used in SAXS/AUC should be discussed in the main text and figure captions in the same units (μM).

Response: We've changed the units used in SAXS to μM .

4. Regarding AUC, I agree that peak broadening suggests that there is a fast exchange. If that is the case, could 1.4 S still be a lower bound for the sedimentation coefficient of the complex?

Response: We appreciate the reviewer's comments. No, 1.4 S is not a lower bound for EGCG-NTD complex, rather it corresponds to the compact form of EGCG-NTD complex. EGCG-NTD complex can also exists in an extended form, with smaller sedimentation coefficient.